# Attenuated palmitoylation of serotonin receptor 5-HT1A affects receptor function and contributes to depression-like behaviors

Nataliya Gorinski [1], Monika Bijata[1,2], Sonal Prasad[1], Alexander Wirth[1], Dalia Abdel Galil[1], Andre Zeug [1], Daria Bazovkina[3], Elena Kondaurova[3], Elizabeth Kulikova[3], Tatiana Ilchibaeva[3], Monika Zareba-Koziol [2], Francesco Papaleo [4], Diego Scheggia [4], Gaga Kochlamazashvili[4], Alexander Dityatev [4,5], Ian Smyth[6], Adam Krzystyniak[2], Jakub Wlodarczyk[2], Diethelm W. Richter[7], Tatyana Strekalova[8,9,10], Stephan Sigrist[11], Claudia Bang[12], Lisa Hobuß[12], Jan Fiedler[12], Thomas Thum [12], Vladimir S. Naumenko[3,14], Ghanshyam Pandey[13,14] & Evgeni Ponimaskin [1,14]

The serotonergic system and in particular serotonin 1A receptor (5-HT1AR) are implicated in major depressive disorder (MDD). Here we demonstrated that 5-HT1AR is palmitoylated in human and rodent brains, and identified ZDHHC21 as a major palmitoyl acyltransferase, whose depletion reduced palmitoylation and consequently signaling functions of 5-HT1AR. Two rodent models for depression-like behavior show reduced brain ZDHHC21 expression and attenuated 5-HT1AR palmitoylation. Moreover, selective knock-down of ZDHHC21 in the murine forebrain induced depression-like behavior. We also identified the microRNA miR-30e as a negative regulator of *Zdhhc21* expression. Through analysis of the post-mortem brain samples in individuals with MDD that died by suicide we find that miR-30e expression is increased, while ZDHHC21 expression, as well as palmitoylation of 5-HT1AR, are reduced within the prefrontal cortex. Our study suggests that downregulation of 5-HT1AR palmitoylation is a mechanism involved in depression, making the restoration of 5-HT1AR palmitoylation a promising clinical strategy for the treatment of MDD.

[1] Cellular Neurophysiology, Hannover Medical School, Carl-Neuberg Str. 1, 30625 Hannover, Germany. [2] Cell Biophysics, Nencki Institute, Pasteur Str. 3, 02-093 Warsaw, Poland. [3] Behavioural Neurogenomics, Institute of Cytology and Genetics, Novosibirsk 630090, Russia. [4] Neuroscience and Brain Technologies, Istituto Italiano di Tecnologia, 16163 Genova, Italy. [5] Molecular Neuroplasticity, DZNE, Leipziger Str. 44, 39120 Magdeburg, Germany. [6] Anatomy & Developmental Biology, Monash University, 3800 Melbourne, Australia. [7] Neuro and Sensory Physiology, University of Göttingen, 37073 Göttingen, Germany. [8] Sechenov First Moscow State Medical University, Moscow, Russia. [9] Neuroscience, Maastricht University, 6229 ER Maastricht, Netherlands. [10] Laboratory of Psychiatric Neurobiology and Institute of General Pathology and Pathophysiology, Sechenov First Moscow State Medical University, Trubetskaya 8, 119315 Moscow, Russia. [11] Institute of Biology, Free University Berlin, Takustr. 6, 14195 Berlin, Germany. [12] Institute of Molecular and Translational Therapeutic Strategies, Hannover Medical School, Carl-Neuberg Str. 1, 30625 Hannover, Germany. [13] Department of Psychiatry, University of Illinois, 1601 W. Taylor Street, Chicago, IL 60612, USA. [14] These authors contributed equally: Vladimir S. Naumenko, Ghanshyam Pandey, Evgeni Ponimaskin. Correspondence and requests for materials should be addressed to V.S.N. (email: naumenko2002@mail.ru) or to G.P. (email: GPandey@psych.uic.edu) or to E.P. (email: Ponimaskin.evgeni@mh-hannover.de)

Major depressive disorder (MDD) is a prevalent mental illness associated with mortality and secondary morbidity. MDD often leads to suicide, with approximately 800,000 people who die are due to suicide every year[1]. Dysregulation of the serotonin system has long been considered central to the etiology of MDD, and the main inhibitory serotonin receptor 5-HT1A (5-HT1AR) seems to play a key role in depressive neuropathology[2,3] (Supplementary Fig. 1A). 5-HT1AR is involved in the regulation of states of depression and anxiety[4]. It has been shown that 5-HT1AR-deficient mice demonstrate increased anxiety-related responses, fear conditioning, and increased freezing behavior[5]. The overexpression of the 5-HT1AR induced during the early postnatal period in the forebrain, but not in the raphe nuclei, has been found to be sufficient to rescue the behavioral phenotype of the knockout mice. These data suggest an important role of postsynaptic 5-HT1AR in psychiatric disorders[6]. It has been also demonstrated that decreased level of 5-HT1AR can lead to anxiety and stress disorders in mice and primates[7,8]. Despite much effort, the molecular routes impairing the serotonin system in clinical depression and suicide remain largely enigmatic.

The 5-HT1AR couples to a variety of effectors via pertussis toxin-sensitive heterotrimeric G proteins of the $G_{i/o}$ family, leading to the agonist-promoted adenylyl cyclase inhibition and a subsequent decrease of the cAMP level[9]. In addition, G-protein gated inwardly rectifying potassium (GIRK or Kir3) channels constitute a central physiological downstream target of 5-HT1AR in the brain (Supplementary Fig. 1A), and cultured neurons produce a robust $K^+$ current after 5-HT1AR stimulation via GIRK channels[10,11]. Previously, we showed that 5-HT1AR is palmitoylated at its C-terminal cysteine residues Cys417 and Cys420 and that the mutation of palmitoylated cysteines reduces $G_i$-protein receptor-mediated signaling via mis-localization of receptor outside of the lipid rafts[12,13]. We also found that 5-HT1AR palmitoylation efficiency was not modulated by the receptor stimulation with agonists[13].

In the present study, we found the palmitoylation of 5-HT1AR to be specifically reduced within the prefrontal cortex (PFC) in the postmortem samples from individuals with MDD who died by suicide (DS). We also identified ZDHHC21 as the cognate palmitoylating enzyme and found that reduced ZDHHC21 expression in the PFC of suicide subjects was connected to attenuated 5-HT1AR palmitoylation status. These findings were mirrored in several rodent models for depression-like behavior. Searching for the mechanisms tuning ZDHHC21 expression, we identified microRNA (miRNA) miR-30e as an important expression regulator and found miR-30e expression to be significantly increased within the PFC of the postmortem samples from suicide individuals with MDD who DS.

## Results

### ZDHHC21 is a major 5-HT1AR palmitoyl acyltransferase.
Using a modified protocol of the acyl-biotinyl exchange (ABE) approach[14] (Supplementary Fig. 1B), we demonstrated palmitoylation of endogenous 5-HT1A receptors in rodent and human brains (Fig. 1a), which was consistent with our previous finding that recombinant 5-HT1AR is palmitoylated[12]. To identify the enzyme(s) palmitoylating the 5-HT1AR, we co-transfected each of the 23 hemagglutinin (HA)-tagged mouse ZDHHC acyltransferases and 5-HT1AR-CFP into neuroblastoma N1E cells metabolically labeled with $[9,10(n)^3H]$-palmitic acid. Substantially increased 5-HT1AR palmitoylation was observed after co-expression of ZDHHC5, -9, and -21 (Supplementary Fig. 2A, B). Role for these ZDHHCs was further confirmed by the ABE approach, with ZDHHC21 as the only acyl-transferase that

significantly increased 5-HT1AR palmitoylation (Fig. 1b, c). Notably, the candidate ZDHHCs belong to the three different subfamilies of ZDHHC proteins with a relative low amino acid sequence homology. Sequence homology between human ZDHHC5 and ZDHHC9 is 43%, between ZDHHC21 and ZDHHC9, 29%, and between ZDHHC21 and ZDHHC5, 26%. Percentages of homology in mouse and rat are quite similar. On the other hand, each of these ZDHHCs is highly conserved between different mammalian species (with approximately 98% homology between mouse, rat, and human isoforms). The three candidate ZDHHCs show a different intracellular distribution, with ZDHHC5 residing at the plasma membrane and ZDHHC9 and -21 localized to the Golgi[15] (Supplementary Fig. 2C–E). Noteworthy, overexpression of all relevant ZDHHCs does not affect a preferential plasma membrane localization of the 5-HT1AR (Supplementary Fig. 2D, E). All these ZDHHCs appeared co-localized with 5-HT1AR. Direct evidence for interaction between 5-HT1AR and ZDHHC5, -9, and -21 was provided in co-immunoprecipitation experiments (Fig. 1d). In these experiments, specific interaction between 5-HT1AR and relevant ZDHHCs was analyzed by co-immunoprecipitation experiments in N1E-115 cells co-expressing HA-tagged 5-HT1AR and green fluorescent protein (GFP)-tagged ZDHHCs. Figure 1d shows that, after immunoprecipitation (IP) with an antibody against the GFP-tag, the HA-tagged receptor could be identified only in samples derived from cells co-expressing both HA- and GFP-tagged proteins. To determine artificial protein aggregation, cells that expressed only one of the proteins (either HA-5-HT1AR or GFP-tagged ZDHHC) were mixed prior to lysis and analyzed in parallel ("mix" samples). As shown in Fig. 1d, both 5-HT1AR and ZDHHCs can be detected by the corresponding antibody (visible in "input" fraction), but no co-immunoprecipitation was observed. This further verifies the specificity of 5-HT1AR–ZDHHC interaction.

Silencing of endogenous ZDHHC5, -9, and -21 by specific short hairpin RNAs (shRNAs, Supplementary Fig. 3A, B) decreased palmitoylation of 5-HT1AR without influencing the expression and distribution of 5-HT1AR in N1E cells (Fig. 1e, f; Supplementary Fig. 3C, D). Noteworthy, knockdown of ZDHHC9 and -21 gave the more prominent reduction of 5-HT1AR palmitoylation in comparison to the effect mediated by shRNA against ZDHHC5 (Fig. 1f), suggesting that both ZDHHC9 and ZDHHC21 represent relevant palmitoylacyltransferases (PATs) for 5-HT1AR. Based on the results obtained after ZDHHC overexpression (Fig. 1b, c), we decided to focus on ZDHHC21 as a more potent PAT for 5-HT1AR. To determine the importance of ZDHHC21 for 5-HT1AR palmitoylation in vivo, we used a ZDHHC21-deficient mouse model, $Zdhhc21^{dep/dep}$. The genetic background of this mouse includes a spontaneous 3-bp deletion in the coding region of the $Zdhhc21$ gene, resulting in non-functional ZDHHC21[16]. In the brains of newborn $Zdhhc21^{dep/dep}$ mice (P0), palmitoylation of 5-HT1AR was significantly impaired (Fig. 1g), while global palmitoylation profiles as well as palmitoylation of the NCAM140 protein, a known ZDHHC3 substrate[17], were not affected (Supplementary Fig. 3F–H). It is noteworthy that, in contrast to results obtained in newborn animals, we did not observe any decrease in palmitoylation of 5-HT1AR in the brains of adult (P30) $Zdhhc21^{dep/dep}$ mice, demonstrating strong compensatory effect during development (Supplementary Fig. 3E).

### ZDHHC5, -9, and -21 regulate 5-HT1AR-mediated signaling.
We have previously demonstrated that non-palmitoylated mutant of 5-HT1AR possesses impaired signaling properties[12,13]. Therefore, we next analyzed whether knocking down the cognate

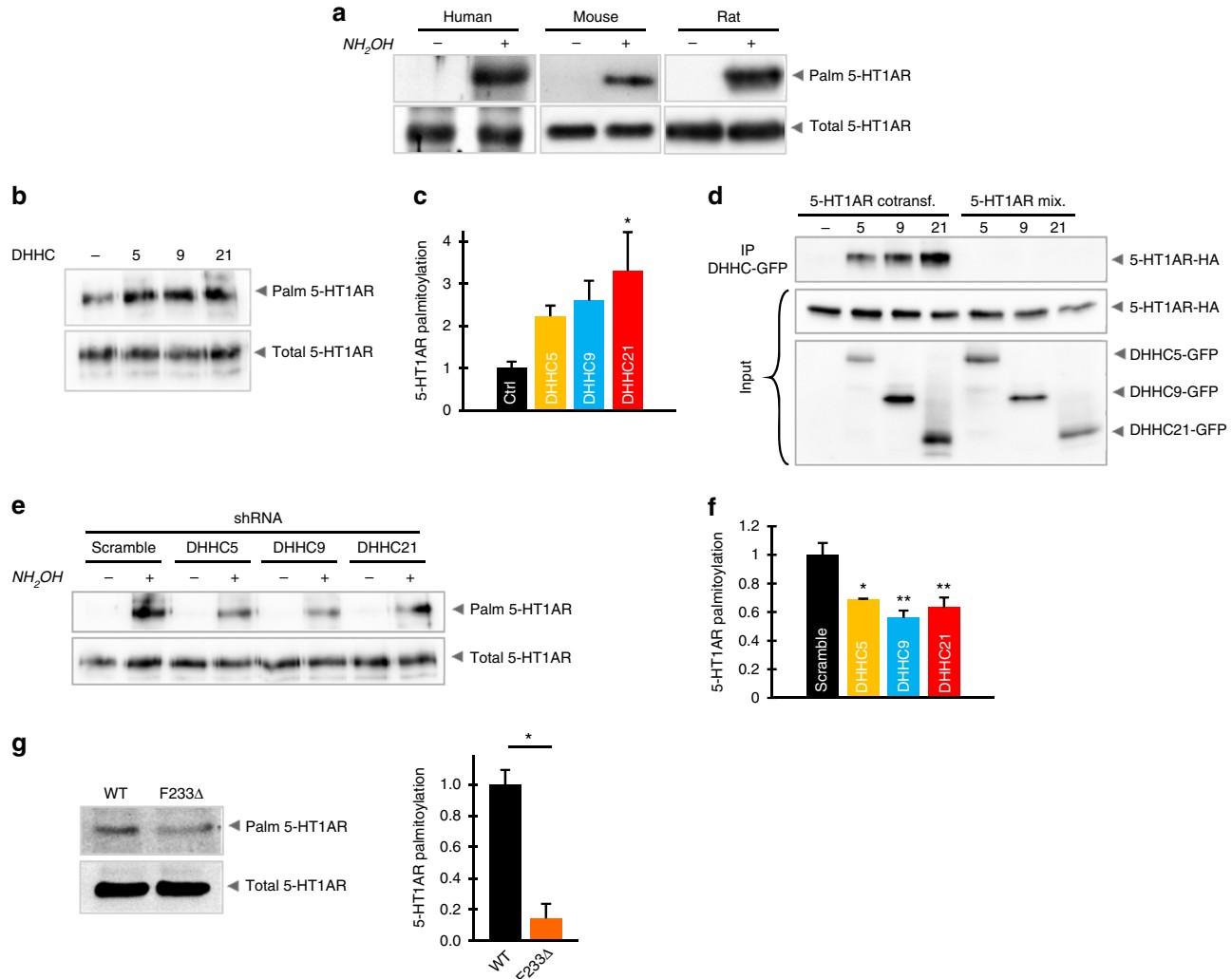

**Fig. 1** Endogenous 5-HT1AR is palmitoylated in rodent and human brains and ZDHHC21 is a major receptor palmitoyl acyltransferase. **a** Palmitoylation of endogenous 5-HT1AR in human, mouse, and rat brains was assessed by the acyl-biotinyl exchange (ABE) assay (see also Supplementary Fig. 1B). **b** Neuroblastoma N1E cells co-expressing 5-HT1AR and indicated ZDHHCs were analyzed by ABE (see also Supplementary Fig. 2A, B). **c** Quantification of 5-HT1AR palmitoylation after co-expression of ZDHHC5, -9, and -21 ($n = 5$). **d** N1E cells were co-transfected with hemagglutinin (HA)-tagged 5-HT1AR and indicated green fluorescent protein (GFP)-tagged ZDHHCs, followed by immunoprecipitation (IP) with anti-GFP antibody and western blot with anti-HA antibody. As a control, mixed lysates from the singly transfected cells (mix) were applied to IP. **e** Analysis of 5-HT1AR palmitoylation in N1E cells after knockdown of endogenous ZDHHC5, -9, and -21 by short hairpin RNA (see also Supplementary Fig. 3A–E) together with **f** quantification. Western blots shown in **e** are representative of at least four independent experiments. **g** Brain tissues isolated from the newborn (P0) F233Δ Zdhhc21$^{dep/dep}$ ($n = 7$) and wild-type ($n = 6$) mice were collected for ABE analysis followed by quantification (see also Supplementary Fig. 3E–H). Bars show means ± SEM; *$P <$ 0.05; **$P <$ 0.01; one-way analysis of variance for **c**, **f**; two-tailed $t$ test for **g**. Source data are available as a Source Data file

5-HT1AR ZDHHCs would affect the receptor functions. Analysis of 5-HT1AR-mediated cAMP changes in single N1E cells using a fluorescence resonance energy transfer-based biosensor CEPAC[18] revealed a quick and strong inhibition of forskolin-evoked cAMP elevation upon stimulation of the wild-type 5-HT1AR. In contrast, expression of the palmitoylation-deficient C417/420S mutant resulted in a significant slowdown of cAMP response kinetics and decrease in response amplitude (Fig. 2a, c, Supplementary Fig. 4A–C). A prominent attenuation of receptor-mediated cAMP response was also observed after knocking down of endogenous ZDHHC5, -9, or -21 in cells expressing 5-HT1AR (Fig. 2a, c).

The 5-HT1AR is involved in activation of the mitogen-activated protein kinase (MAPK) extracellular signal–regulated kinase 2 (Erk2) either by a G-protein-independent pathway or via

$G_{\beta\gamma}$ subunits[9], while the ability of palmitoylation-deficient 5-HT1AR to activate this pathway is substantially reduced[12] (Fig. 2d, e). Analysis of Erk phosphorylation after receptor stimulation revealed that such activation was significantly decreased only after knocking down of ZDHHC9 and -21, suggesting that ZDHHC5, which is localized at the plasma membrane, is not involved in this pathway (Fig. 2d, e, Supplementary Fig. 4D, E). The 5-HT1AR (C417/420S) mutant here served as a negative control.

Silencing ZDHHC21 in cultured hippocampal neurons diminished both basal as well as 5-HT1AR agonist (i.e., 8-OH-DPAT)-evoked GIRK currents, when compared with scramble shRNA controls (Fig. 2f, h). These combined results demonstrate that ZDHHC5 and -9, and in particular ZDHHC21, can regulate 5-HT1AR-mediated signaling via modulating the receptor palmitoylation state.

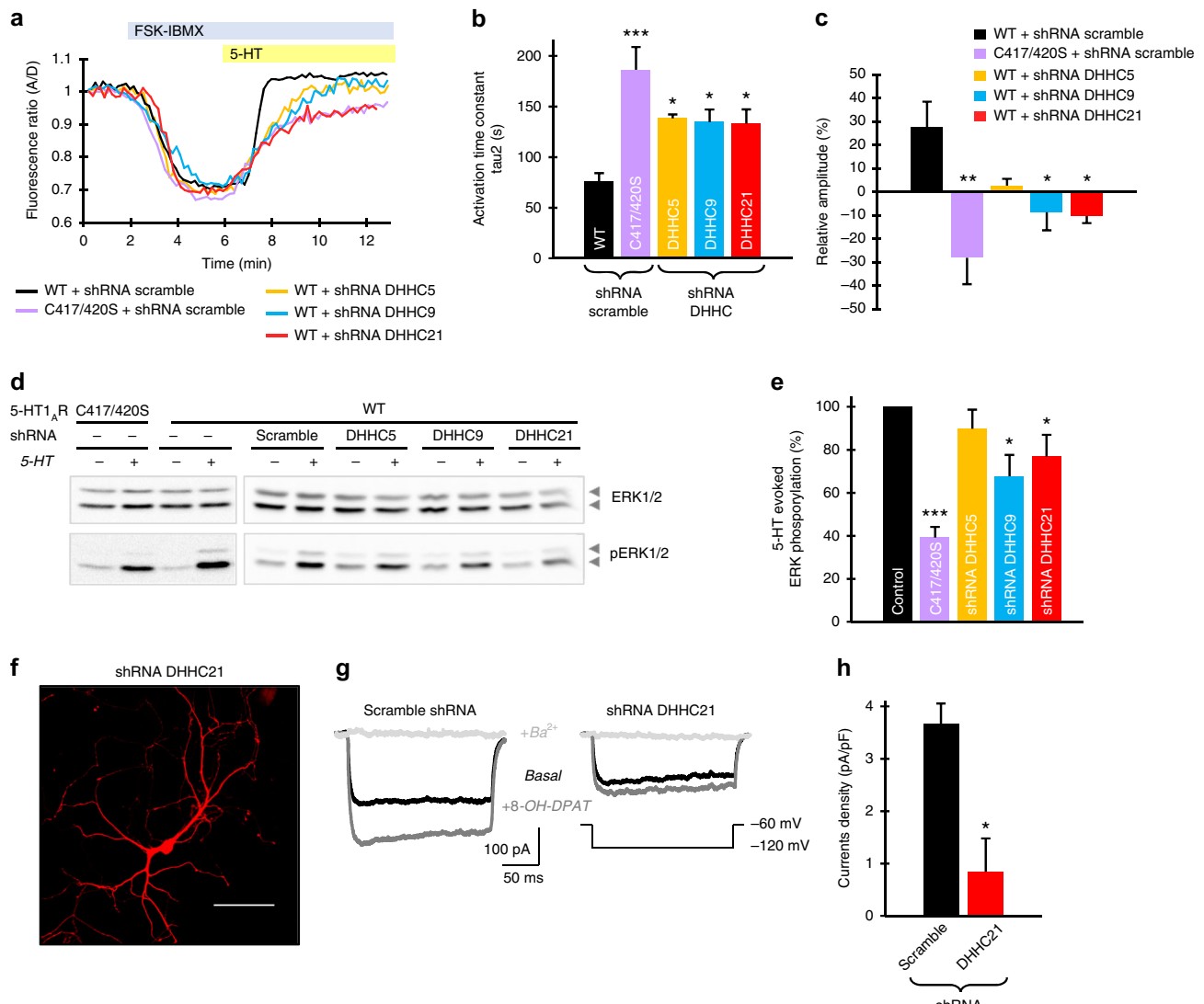

**Fig. 2** Knockdown of ZDHHC5, -9, and -21 impairs 5-HT1AR-mediated signaling. **a** N1E cells were transfected with cAMP fluorescence resonance energy transfer-based biosensor CEPAC and 5-HT1AR-mCherry along with the indicated constructs (see also Supplementary Fig. 4A–C). After pretreatment with 1 μM forskolin and 25 μM 3-isobutyl-1-methylxanthine, cells were stimulated with 20 μM serotonin (5-HT). Each trace shows cAMP response at the single cell. **b** Graphs show activation time constant and **c** changes of cAMP response amplitude relative to pretreatment ($N = 4$, in each experiments at least 20 cells were analyzed; one-way analysis of variance (ANOVA)). **d** Analysis of extracellular signal–regulated kinase (Erk) phosphorylation in N1E cells expressing the indicated constructs after stimulation with 10 μM 5-HT. **e** Quantification of 5-HT evoked Erk phosphorylation calculated as the ratio of total Erk expression over the Erk phosphorylation signal. Bars show means ± SEM ($N \geq 7$, one-way ANOVA; see also Supplementary Fig. 4D, E). **f** Representative image of hippocampal neuron expressing short hairpin RNA against ZDHHC21 together with red fluorescent protein at the day in vitro 11. Scale bar: 100 μm. **g** Examples of G-protein gated inwardly rectifying potassium (GIRK) channel currents in two transfected groups after application of 5-HT1AR agonist 8-OH-DPAT and channel blocker BaCl2. **h** A summary of recordings from three independent culture preparations. Bars show means ± SEM of the amplitude of 8-OH-DPAT-stimulated GIRK currents. *$P < 0.05$; **$P < 0.01$; ***$P < 0.001$, two-tailed $t$ test. Source data are available as a Source Data file

**Palmitoylation of 5-HT1AR in rodent depression models.** Using two different rodent models, we next investigated whether the decreased 5-HT1AR palmitoylation can be associated with depressive-like behavior. We first evaluated 5-HT1AR palmitoylation in a paradigm of stress-induced anhedonia (decreased reward sensitivity), serving as a well-established mouse model for depression-like behavior[19–21]. After chronic application of three stressors — social defeat stress, restraint stress, and tail suspension stress — mice exhibiting a sucrose preference of <65% were defined as anhedonic, whereas mice that maintained a sucrose preference of more than >65% were defined as resilient to a depressive-like syndrome (Fig. 3a, b). Consistent with previous work[19], the immobility times in the forced swimming test were

significantly elevated in anhedonic mice versus the control and resilient groups (Fig. 3c). Analysis of 5-HT1AR palmitoylation in the different brain areas revealed a significant reduction in 5-HT1AR palmitoylation but not in the expression in the PFC, a brain area critically involved in the pathogenesis of depressive symptoms[22,23] and not in the cerebellum and hippocampus of stress-induced anhedonic mice in comparison to the resilient animals (Fig. 3d, e, Supplementary Fig. 5A). Importantly, the global palmitoylation patterns and palmitoylation level of NCAM140 in PFC, hippocampus, and cerebellum were not affected (Supplementary Fig. 5B–E).

The decreased palmitoylation can be mediated via increased activity of the acyl-protein thioesterases, the palmitoyl-protein

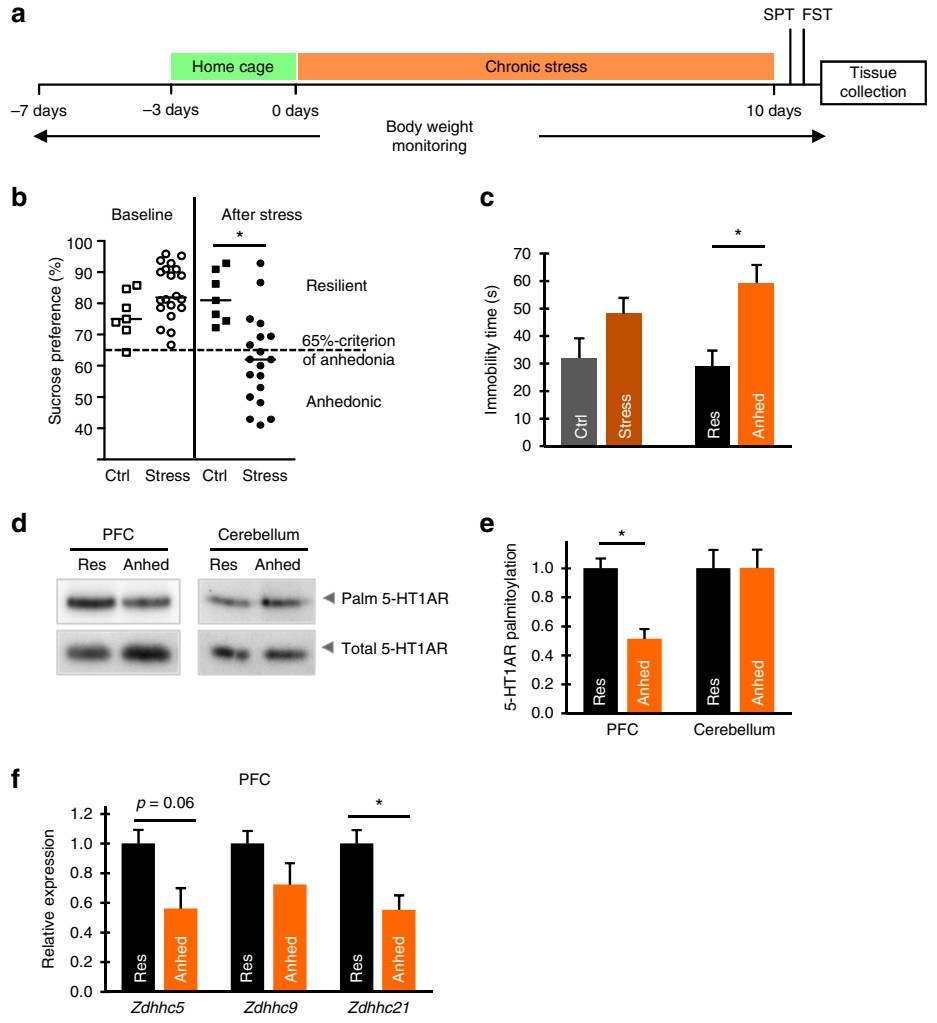

**Fig. 3** Palmitoylation of 5-HT1AR and expression of ZDHHC21 are decreased in the forebrains of mice with depressive-like syndrome. **a** Schematic diagram of experimental design for mouse model of depression. **b** Results of the sucrose preference test performed in control ($n = 7$) and chronically stressed mice ($n = 19$). **c** Immobility time in the forced swim test of control and chronically stressed mice. Group of stressed mice was then divided into anhedonic (anhed) ($n = 12$) and resilient (res) ($n = 7$) animals and immobility time was analyzed in each subgroup. One-way ANOVA. Data points represent mean ± SEM (*$P < 0.05$). **d** In anhedonic and resilient mice, palmitoylation of the 5-HT1AR was analyzed in the prefrontal cortex (PFC) and cerebellum by acyl-biotinyl exchange (see also Supplementary Fig. 5A–E). **e** Quantification of 5-HT1AR palmitoylation in anhedonic and resilient animals (see also Supplementary Fig. 5A–E). **f** Relative expression levels of ZDHHC5, -9, and -21 in the PFC of resilient and anhedonic mice assessed by quantitative reverse transcriptase-PCR. *$P < 0.05$; two-tailed $t$ test. (See also Supplementary Fig. 5F–H). Source data are available as a Source Data file

thioesterases[24], or serine hydrolases (so-called alpha-beta hydro-lase domain or ABHD proteins) with depalmitoylating activity[25,26]. Alternatively, reduced 5-HT1AR palmitoylation could be elicited by decreased activity and/or expression of corresponding ZDHHCs. The latter is more likely the case for 5-HT1AR because its palmitoylation is a stable modification with a lifetime corresponding to the lifetime of the receptor itself[12]. Therefore, we investigated whether the expression level of ZDHHC5, -9, and -21 is affected in the brain of anhedonic mice. All three ZDHHCs are known to be ubiquitously expressed in the brain[27,28], and we found homogeneous distribution of these ZDHHCs throughout the mouse brain with high levels of ZDHHC9 and -21 expression in the cortex (Supplementary Fig. 5F). Quantitative reverse transcriptase PCR (RT-PCR) analysis revealed that the *Zdhhc21* expression was significantly attenuated only in the PFC of anhedonic mice when compared to the resilient animals (Fig. 3f; Supplementary Fig. 5G, H).

To independently validate these results, we applied a restraint stress-based depression model in rats[29] (Fig. 4a). After 3 weeks of

being restrained for 6 h per day, animals developed a strong depression-like phenotype characterized by increased corticoster-one levels, lower weight gain, and increased cumulative duration of immobility in the force swimming test (Fig. 4b, e). Similar to results obtained in anhedonic mice, only palmitoylation and not the expression of 5-HT1AR in the PFC, but not in the cerebellum or hippocampus, of stressed rats was significantly decreased without changes in global palmitoylation or in palmitoylation of NCAM140 (Fig. 4f, g; Supplementary Fig. 6A–E). Consistent with the finding in mouse depression model, *Zdhhc21* expression level was also reduced in the PFC of rats with depression-like behavior (Fig. 4h). Interestingly, in the rat model for depressive-like behavior, we found an additional decrease in the amount of *Zdhhc5* and *Zdhhc9* transcripts in the PFC, while the expression of *Zdhhc5, 9* and *21* in the hippocampus and cerebellum was not affected (Fig. 4h; Supplementary Fig. 6F, G).

Collectively, our rodent models data suggest attenuated expression level of ZDHHC21 and reduced 5-HT1AR palmitoy-lation as a general signature for depressive-like behavior.

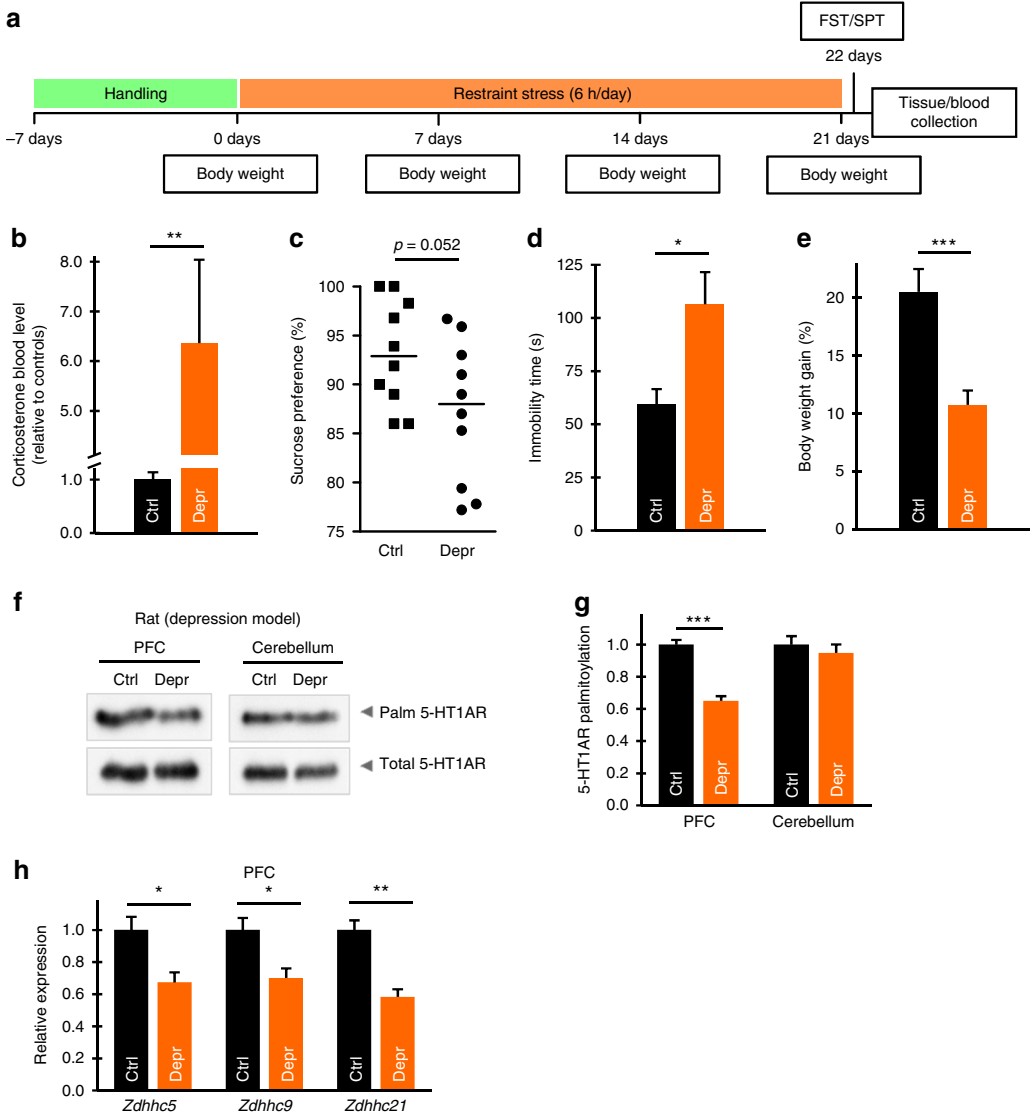

**Fig. 4** Palmitoylation of 5-HT1AR and expression of ZDHHC21 are decreased in the forebrains of rats with depression-like phenotype. **a** Schematic diagram of experimental design for restraint stress depression model in Wistar rats. **b** Normalized corticosterone blood level of the control (ctrl) and rats with depression-like behavior (depr). **c** Results of the sucrose preference test in control and chronically stressed rats. **d** The duration of immobility in the forced swim test of the control and rats with depression-like behavior. **e** Body weight gain during experiment of the control and rats with depression-like behavior. Data points in **b**–**e** represent mean ± SEM. ($n = 10$ for each group; *$P < 0.05$; **$P < 0.01$; ***$P < 0.001$). **f** Prefrontal cortex (PFC) and cerebellum of the control Wistar rats (ctrl) and rats with depression-like behavior (depr) were subjected to acyl-biotinyl exchange to visualize 5-HT1AR palmitoylation. **g** Bar graphs representing normalized level of 5-HT1AR palmitoylation in control ($n = 5$) and rats with depression-like behavior ($n = 5$) animals. All blots are representative of at least three independent experiments. Data points represent the means ± SEM. *$P < 0.05$; ***$P < 0.001$, two-tailed $t$ test (see also Supplementary Fig. 6A–E). **h** Relative expression levels of ZDHHC5, -9, and -21 in the PFC of control rats and rats with depression-like phenotype (two-tailed $t$ test, see also Supplementary Fig. 6F, G). Source data are available as a Source Data file

**ZDHHC21 knockdown in PFC triggers depressive symptoms.** To test for a causal relationship among the level of ZDHHC21, the degree of 5-HT1AR palmitoylation, and initiation of depressive symptoms, we selectively knocked down the expression of this palmitoyl acyltransferase in the PFC of mice. To do so, we generated an adenovirus-associated viral (AAV) construct encoding the shRNAs to silence endogenously expressed ZDHHC21 along with the red fluorescent protein under control of the synapsin promoter, which allowed for visualization of the infected neurons (Fig. 5a; Supplementary Fig. 7A–C, E). Bilateral administration of this construct into the PFC of C57BL/6J mice attenuated the expression of *Zdhhc21* in the forebrain when compared to mice treated with vehicle or scrambled shRNA (Fig. 5b). Of note, the cortical AAV injection did not modulate

*Zdhhc21* expression in the non-injected brain regions, like cerebellum (Supplementary Fig. 7F). Also, the expression of 5-HT1AR in the PFC, hippocampus, and cerebellum was not changed after AAV injection (Fig. 5c, e; Supplementary Fig. 7G). General palmitoylation patterns as well as palmitoylation of NCAM140, used as a control, were not affected in the mouse brain after PFC injection of AAV encoding for the shRNA against ZDHHC21 (Supplementary Fig. 7H, I). In contrast, 5-HT1AR palmitoylation but not the receptor expression was selectively reduced in the PFC (Fig 5c, e, Supplementary Fig. 7G), further confirming ZDHHC21 as a main 5-HT1AR palmitoyl acyltransferase. Of importance, PFC-specific ZDHHC21 knockdown triggered depression-like behavioral traits. In particular, the immobility time in the forced swim test (FST) used as a readout

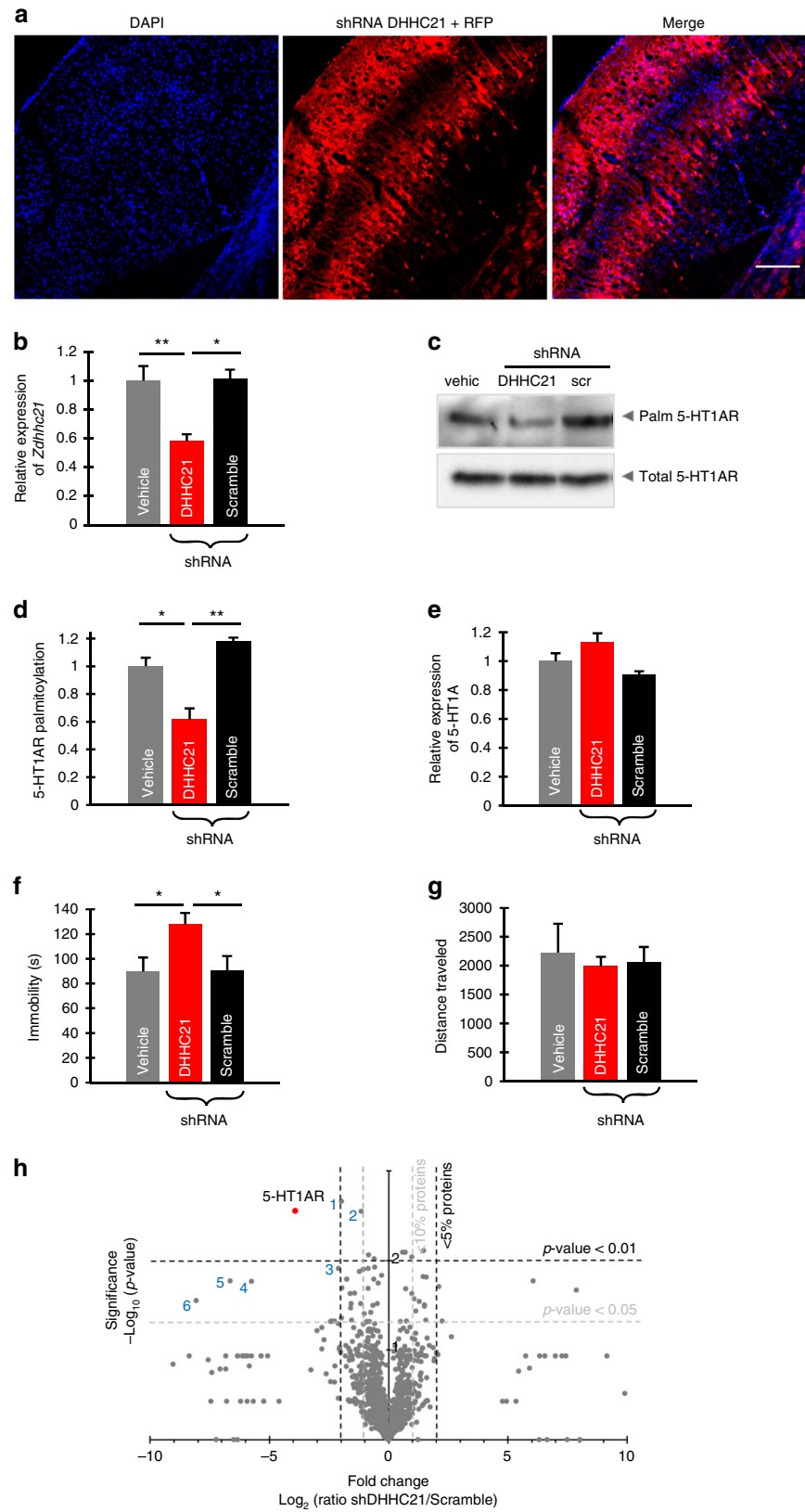

for depressive-like behavior was significantly increased (Fig. 5f), without affecting locomotor, anxiolytic, or exploratory activity as assessed by the open field test (Fig. 5g, Supplementary Fig. 7J, K). In addition, we did not obtain any differences between groups in the tail suspension and the novel object recognition tests (Supplementary Fig. 7L, M). The lack of visible effect in the tail suspension test, which is often used for analysis of the depression-like behavior in rodent, can be explained by the fact that the majority of the C57BL/6J mice tested in this paradigm climbed up their tails during the test session. Such behavior is specific for the

**Fig. 5** Selective knockdown of ZDHHC21 in the prefrontal cortex (PFC) leads to development of depressive-like behavior in mice. **a** Cortical slice of the C57BL/6J mouse 30 days after administration of adenovirus-associated virus encoding for short hairpin RNA (shRNA) against ZDHHC21 along with red fluorescent protein (see also Supplementary Fig. 7A, B, E). Scale bar: 200 μm. **b** Relative expression levels of *Zdhhc21* in the PFC after injection of the indicated constructs ($n = 4$ for each group, one-way analysis of variance (ANOVA)). **c** PFCs of mice were isolated 30 days after injection either with vehicle or scrambled shRNA or shRNA against ZDHHC21 and subjected to acyl-biotinyl exchange assay to define 5-HT1AR palmitoylation. Quantification of palmitoylation (**d**) and expression (**e**) of 5-HT1AR in the PFCs of mice after injection either with vehicle or scrambled shRNA or shRNA against ZDHHC21. **f** Immobility time in the forced swim test of mice treated with vehicle ($n = 10$), shRNA against ZDHHC21 ($n = 9$), or scrambled shRNA ($n = 11$). Statistical significance between values is noted (*$P < 0.05$, **$P < 0.01$; one-way ANOVA; see also Supplementary Fig. 7J–M). **g** Comparison of the motor activity assessed by the open field test in mice treated with vehicle, shRNA against ZDHHC21, or scrambled shRNA. **h** Volcano plot depicting differential S-palmitoylation analysis between mice injected with scramble shRNA and with shRNA against ZDHHC21 using mass spectrometry spectral counts data with estimated fold changes ($x$ axis) versus the significance, i.e., $-\log_{10}$ ($P$ values) for each protein ($y$ axis). Dotted vertical lines lineate events with fold change >2. Dotted horizontal lines denote point at which two-tailed $t$ test $P > 0.05$. Value for 5-HT1AR is marked in red ($n = 3$ biological replicates for each condition; see also Supplementary Data 1 and 2). 1, TRPC channel subfamily V (TRPV2); 2, phosphoglycerate mutase 1 (PGAM1); 3, proteosomal ubiquitin receptor ADRM1; 4, exopolyphosphatase PRUNE1; 5, signal transducer and activator of transcription 6 (STAT6); 6, Hsc70-interacting protein. Source data are available as a Source Data file

C57BL/6J mouse line, which is in accordance with previous observations[30]. Thus attenuated palmitoylation of the 5-HT1AR in PFC mediated by decreased ZDHHC21 expression was sufficient for the development of depressive symptoms.

**5-HT1AR is a main substrate for ZDHHC21 in the mouse PFC**. To verify the substrate specificity of ZDHHC21 in the PFC of mouse brain, we applied the quantitative palmitoylomics approach[31]. In particular, we used a high-throughput ABE proteomics approach that enables identification of S-Palmitoylated-Cys sites in complex biological mixtures. Using this method in combination with the mass spectrometry (MS) protein identification to precisely ascertain the targets of S-palmitoylation, we compared palmitoylation profile in the PFC of control animals and those injected with scramble or shRNA against ZDHHC21. Inclusion criteria used for selection of differential protein sets are specified in detail in "Methods" section. For data evaluation, we applied analytic approach developed for the global analysis of proteomics data[32].

Using three biological replicates, we identified 1737 palmitoylated protein in the mouse PFC (Supplementary Data 1 and 2). Detailed analysis of the palmitoylomics results revealed that knockdown of ZDHHC21 provoked a strong (approximately four-fold change plotted in log$_2$ scale, which correspond to approximately 15 times decrease in the palmitoylation level) and highly significant ($P = 0.0028$) decrease of 5-HT1AR palmitoylation in comparison to the *scr* samples (Fig. 5h). Noteworthy that, among three additional proteins, which palmitoylation was reduced to the similar extent, reduction of 5-HT1AR palmitoylation was at the highest level of confidence ($P = 0.0028$), followed by proteosomal ubiquitin receptor ADRM1 ($P = 0.012$), exopolyphosphatase PRUNE1 ($P = 0.017$), STAT6 ($P = 0.017$), and Hsc70-interacting protein ($P = 0.028$). For two other proteins, including TRPC channel subfamily V (TRPV2) and phosphoglycerate mutase 1 (PGAM1), the level of confidence was comparable with that of the 5-HT1AR. However, for these proteins we obtained substantially lower decrease in palmitoylation (1.97-fold change for TRPV2 and 1.16-fold change for PGAM1). Interestingly, with the exception of 5-HT1AR, palmitoylation of all the above-mentioned proteins has not been reported before. More importantly, these proteins have not been reliably associated with MDD. Though the above-mentioned proteins have not been associated with depression, their palmitoylation might theoretically have an impact on the depressive phenotype. We therefore performed additional experiments to directly address the role of ZDHHC21-mediated 5-HT1AR palmitoylation in depression. Multiple prior studies evidently show that the acute 8-OH-DPAT injection results in a strong, 5-HT1AR-mediated anti-depressive effects as assessed by

decreased immobility time in the forced swimming test[33,34]. We were able to reproduce this 5-HT1AR-mediated effect of 8-OH-DPAT in control conditions, when ZDHHC21 was normally expressed (Supplementary Fig. 7N). However, acute injection of 8-OH-DPAT failed to induce any anti-depressive effect in animals after knockdown of ZDHHC21 in PFC (Supplementary Fig. 7N). Together with results obtained in palmitoylomics experiments (Fig. 5h), these combined data provide a direct evidence for ZDHHC21-mediated 5-HT1AR palmitoylation playing a central role in the etiology of depression-like phenotype.

**ZDHHC21 expression and 5-HT1AR palmitoylation in human brains**. To validate the role of 5-HT1AR palmitoylation in MDD, we compared levels of 5-HT1AR palmitoylation in postmortem samples from individuals who DS and diagnosed previously with major depression and control subjects. These groups were matched for age (mean age value in the control group was $44.2 \pm 4.1$ years and in the suicide group it was $42.2 \pm 4.3$ years), gender (36% female in the control and 43% in the suicide group), and race (12.5% of African American in both the groups). Although the global palmitoylation profile as well as the palmitoylation of defined neuronal proteins (e.g., NCAM140) was unaffected (Supplementary Fig. 8B–D), palmitoylation of 5-HT1AR in the PFC was significantly reduced in individuals with MDD who DS (Fig. 6a, b). Of note, the level of 5-HT1AR expression in the PFC of suicide subjects was unaffected (Supplementary Fig. 8A). Furthermore, 5-HT1AR palmitoylation in other brain areas, i.e., those areas not central to MDD pathology, appears not to be reduced (Fig. 6a, b).

Expression analysis in human brain samples from depressive suicide subjects revealed that the amount of *ZDHHC9* and *ZDHHC21* transcripts was significantly downregulated in the PFC of individuals with MDD who DS, while the production of both *ZDHHCs* in the cerebellum was similar to that obtained in the control (Fig. 6c). Interestingly, in the rat model of depressive-like behavior, we found an additional decrease in the amount of *Zdhhc5* transcripts in the PFC that was not seen in the human samples from suicide subjects (Figs. 4h and 6c), demonstrating that ZDHHC5 and ZDHHC9 contributed differently to 5-HT1AR palmitoylation in human and rodent brains under pathological conditions.

**miRNA miR-30e regulates the ZDHHC21 expression**. For a deeper look into the molecular mechanisms regulating DHHC activity in vivo, we focused on the possible role of miRNAs. miRNAs are highly conserved, non-coding RNAs that regulate gene expression of target messenger RNAs by binding to the 3' untranslated region[35]. These molecules also known to regulate expression of ZDHHC9 in neuronal populations, leading to

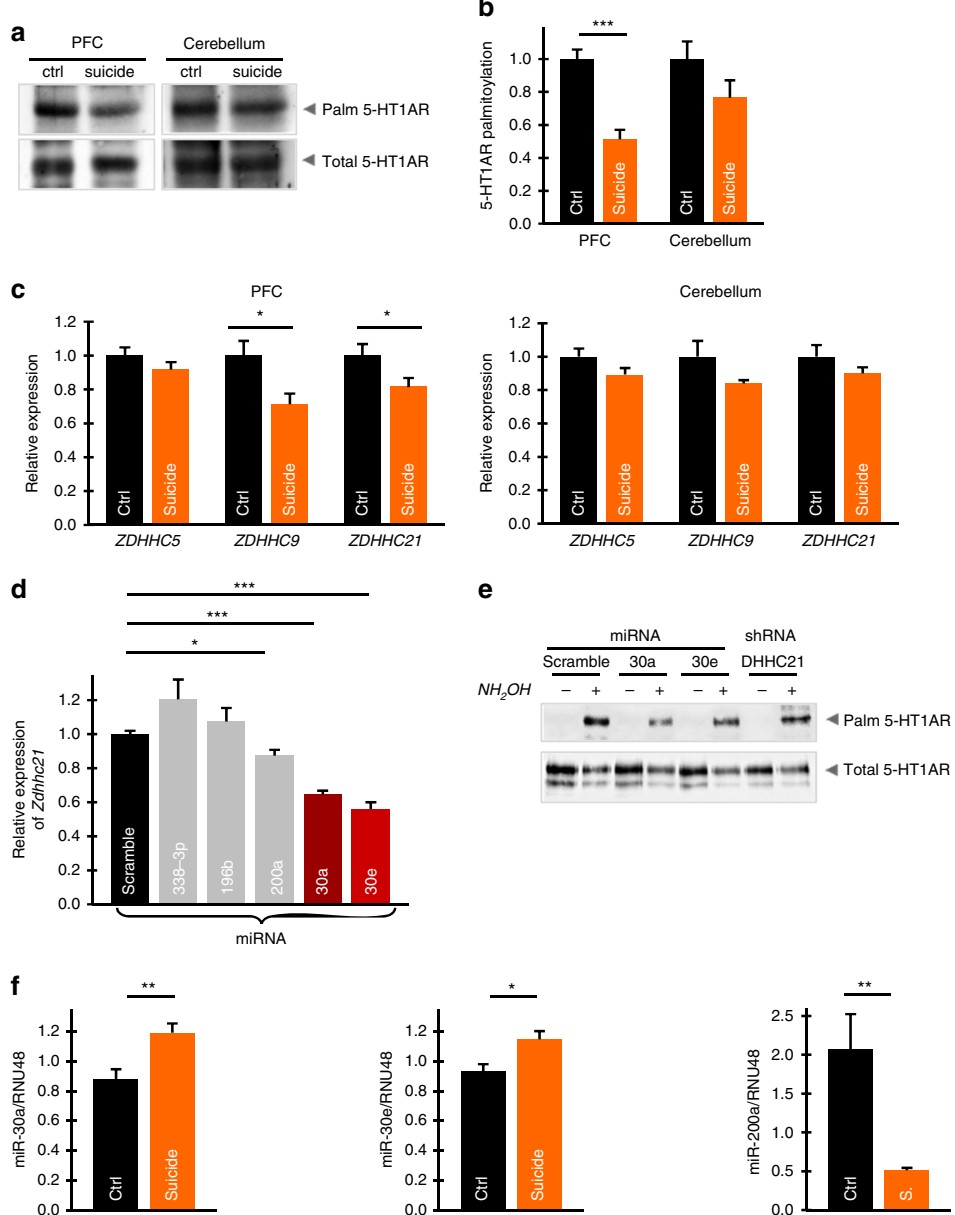

**Fig. 6** Attenuated 5-HT1AR palmitoylation in the prefrontal cortex (PFC) of individuals with major depressive disorder (MDD) who died by suicide correlates with reduced expression of ZDHHC21. **a** 5-HT1AR palmitoylation in the PFC and cerebellum from the control and individuals with MDD who died by suicide (PFC, $n = 5$; cerebellum, $n = 4$). **b** Normalized level of 5-HT1AR palmitoylation in the PFC and cerebellum of individuals with MDD who died by suicide in comparison with control subjects (two-tailed $t$ test; see also Supplementary Fig. 8). **c** Relative expression levels of ZDHHC5, -9, and -21 in the PFC (left) and cerebellum (right) of control ($n \geq 14$) and individuals with MDD who died by suicide ($n \geq 12$). Data points represent the means ± SEM (*$P < 0.05$; ***$P < 0.001$; two-tailed $t$ test). **d** N1E cells were transfected with microRNAs as indicated and the expression of ZDHHC21 was determined by reverse transcriptase-PCR (see also Supplementary Table 1). Data points represent mean ± SEM from at least three independent experiments (*$P < 0.05$; ***$P < 0.001$; two-tailed $t$ test). **e** N1E cells were transfected with 5-HT1AR along with miR-30e, miR30-a, or shRNA against ZDHHC21. Palmitoylation of 5-HT1AR was analyzed by acyl-biotinyl exchange. Western blots shown are representative of at least three independent experiments. **f** Analysis of miR-30a, -30e, and -200a expression in the PFC of control and individuals with MDD who died by suicide. (*$P < 0.05$, **$P < 0.01$; two-tailed $t$ test). Source data are available as a Source Data file

changed localization of its substrate H-Ras[36]. In silico analysis revealed that the 3' untranslated region of the *Zdhhc21* gene contains conserved miRNA-binding sites for miR-338–3p, miR-196b, miR-200a, miR-30a, and miR-30e (Supplementary Table 1). To test the importance of these miRNAs in regulation of ZDHHC21 expression, we mimicked a miRNA environment in neuroblastoma cells by overexpressing the predicted miRNAs and found that transfection of miR-200a, miR-30a, and miR-30e all resulted in significant reductions, while transfection of miR-338-

3p showed slightly increased ZDHHC21 expression (Fig. 6d). We then focused on miR-30a and miR-30e, i.e., the two miRNAs showing the greatest reduction in ZDHHC21 expression, by examining the effects on 5-HT1AR palmitoylation. In these experiments, we found that receptor palmitoylation, but not expression, is reduced with miR-30a and -30e overexpression (Fig. 6e).

To directly evaluate the role of miRNA in MDD, we compared the expression of endogenous miRNA miR-30a, -30e, and -200a in

the postmortem medial PFC (mPFC) samples from individuals with MDD who DS and control subjects. These experiments revealed that the expression of miR-30a and miR-30e was significantly increased, while the expression of miR-200a was drastically reduced in samples from individuals with MDD who DS (Fig. 6f). These results provide a strong experimental support for our hypothesis on the role of defined miRNAs in the development of depression.

## Discussion

The 5-HT1A receptor is a well-studied member of the serotonin receptor family, in part because of its role in regulating depression- and anxiety-like behavior[4]. However, the precise role of 5-HT1AR in the pathogenesis of depressive disorders and treatment of the diseases are not fully understood. Though controversial, several studies suggested a possible relation between depressive disorders and abnormalities in the distribution and/or basal expression of 5-HT1A receptor in the brain. In several studies, decreased 5-HT1A receptor density in the PFC was reported in positron emission tomographic and postmortem studies of MDD patients[2,37,38]. In another study, however, analysis of the postmortem brains of depressed subjects in comparison with control samples revealed a specific upregulation of 5-HT1A autoreceptors in the raphe area, with no changes in postsynaptic 5-HT1AR sites[39]. In contrast, other studies reported that higher 5-HT1AR binding was correlated with increased depressive symptoms and blunted selective serotonin reuptake inhibitor treatment response[40–42]. A possible explanation for such at first glance contradictory observations might be that compromised 5-HT1AR-mediated signaling (e.g., through the mutations within HTR1A gene) rather than changes in the expression level might be responsible for MDD onset. While several rare single-nucleotide polymorphisms (SNPs) within 5-HT1AR gene were described, these mutations have a very low population frequency (<2%) and failed to be associated with mental illness[43–45]. Apart from changes in the expression level and gene mutations, post-translational modifications are often a key for controlling neurotransmitter receptor signaling. Lately, palmitoylation has received increased attention[46].

5-HT1AR has been shown to be palmitoylated at its C-terminal cysteine residues Cys417 and Cys420, which is essential for enabling $G_i$-protein coupling and effector-mediated signaling of this receptor[12,13]. We here demonstrate that a downregulation of 5-HT1AR functions via changes of its palmitoylation status is associated with MDD. We identified ZDHHC5, -9 and -21 as the cognate palmitoylating enzymes and found that reduced ZDHHC21 expression in the PFC of subjects with MDD who DS correlates with attenuated 5-HT1AR palmitoylation status. Most importantly, our experiments proved a causal relationship between the PFC level of ZDHHC21, the degree of 5-HT1AR palmitoylation, and the initiation of depressive symptoms.

It has been shown that depending on site and time of expression defined ZDHHCs can have multiple substrates[47–49]. In the present study, using a comprehensive palmitoylomics approach we thus analyzed the substrate specificity of the more potent PAT for 5-HT1AR, ZDHHC21. Results of our palmitoylomics experiment confirmed 5-HT1AR as a main substrate for ZDHHC21 in the PFC. Although we cannot completely exclude an impact of other proteins, whose palmitoylation was affected under ZDHHC21 knockdown (e.g., PRUNE1, STAT6, Hsc70-interacting protein, TRPV2, PGAM1), these proteins have not been reliably associated with MDD. In combination with results obtained after acute injection of 5-HT1AR agonist 8-OH-DPAT, which failed to evoke any anti-depressive effects in animals after knockdown of DHHC21, this data provides a direct evidence for ZDHHC21-mediated 5-HT1AR palmitoylation playing a central role in the etiology of depressive phenotype.

Noteworthy, even though the substrate specificity represents a critical component determining functional consequences of DHHC activity, this issue was not yet systematically investigated for the other ZDHHC members. In the majority of ZDHHC-related studies, including identification of DHHCs responsible for palmitoylation of phospholemman[50], melanocortin receptor MCR1[51], and CD36[52], the question about the substrate specificity was not addressed in detail.

Searching for mechanisms tuning ZDHHC21 expression, we identified several miRNAs being able to modulate 5-HT1AR palmitoylation. miRNAs are a subclass of small, noncoding, single-stranded RNA molecules responsible for the post-transcriptional repression or degradation of target mRNA[53]. Of note, aberrant expression of specific miRNA is associated with several psychiatric disorders, including bipolar disorder and schizophrenia[54,55]. Several studies also demonstrated altered miRNA expression profile in postmortem as well as peripheral blood samples from MDD patients[56–58]. One of these studies identified polymorphisms within the miR-30e to be positively correlated with depression and its symptomatic onset[59]. MiR-30e, an established tumor suppressor, was also found to be associated with the development of schizophrenia[60,61], although its exact targets have not been validated. Intriguingly, our in silico search for miRNAs that might be able to bind to the 3' untranslated region of the Zdhhc21 gene revealed miR-30e as a candidate for regulation of ZDHHC21 expression. In fact, transfection of miR-30e significantly reduced the expression of Zdhhc21 and consequently 5-HT1AR palmitoylation, suggesting increased expression of miR-30e as an important factor involved in initiation of depressive symptoms. A strong experimental support for our hypothesis on the role of defined miRNAs in the etiology of MDD was provided by the fact that the expression of endogenous miR-30a and miR-30e was significantly increased, while the expression of miR-200a was abolished in samples from individuals with MDD who DS. It would therefore be interesting to evaluate in future studies miR-30a, -30e, and -200a as potential biomarkers to identify MDD patients with a suicidal mind.

Changes in miRNA expression represent an important part of the stress response leading to MDD[62,63]. In search for underlying mechanisms, an increased expression of transcription factor REST4 in PFC was discovered in the maternally separated rat stress model[64]. As REST4 is known to regulate brain-enriched miRNAs[65,66], further studies are warranted to validate whether a stress-induced increase of REST4 expression in the PFC might result in a local upregulation of miRNAs, including miR-30e, which in turn represses ZDHHC21 expression.

In conclusion, our study uncovered a causal link between regulated levels of ZDHHC21 expression, attenuated 5-HT1AR palmitoylation, and development of depressive symptoms making the restoration of 5-HT1AR palmitoylation (either by upregulation of ZDHHC21 or inhibition of miR-30a or -30e) a possible clinical strategy for the treatment of MDD.

## Methods

**Human samples.** Assessment of 5-HT1AR palmitoylation in the human brains was carried out in the PFC (Brodmann area 9) and cerebellum of individuals with MDD who DS and 16 non-psychiatric control subjects, referred to normal controls (NC). Brain tissues were obtained from the Maryland Brain Collection at the Maryland Psychiatric Research Center, Baltimore. Tissues were collected only after a family member gave informed consent. All tissue from NC and DS subjects was examined grossly by experienced neuropathologists. Toxicology data were obtained by the analysis of urine and blood samples. All procedures were approved by the University of Maryland and University of Illinois institutional review boards. Diagnosis of subjects was based on the Structured Clinical Interview for Diagnostic and Statistical Manual of Mental Disorders, Fourth Edition (SCID I)[67]. At least one family member and/or a friend, after giving written informed consent, underwent

an interview. Diagnoses were made by a consensus of two psychiatrists from the data obtained in this interview, medical records from the case, and records obtained from the medical examiner's office. Normal control subjects were verified as free from mental illnesses using these consensus diagnostic procedures.

**Animals.** The behavioral experiments were with 3-month-old male C57BL/6J wild-type mice (mouse model of stress-induced anhedonia) and 3-month-old male Wistar rats (restraint stress model). All procedures performed on animal were according to the guidelines of the European Communities Council Directive for the care and use of laboratory animals and approved by the respective local governmental bodies (permission 0421/000/000/2013 was issued by General Directory of Ethical Committee of the New University of Lisbon, in accordance with Portuguese Law-Decrees DL129/92, DL197/96 and Ordinance Port.131/97 and of the First Warsaw Ethical Committee on animal research (permission no. 554/2013). For the intracortical administration of AAV, adult C57BL/6J male mice (8 weeks old) weighing about $25 \pm 2$ g were used. Mice were housed in the vivarium of the Institute of Cytology and Genetics (Novosibirsk, Russia) under standard laboratory conditions in a natural light–dark cycle (12 h light and 12 h dark) with free access to water and food. Two days before experiments, the mice (15 animals for each of the 3 groups) were isolated in individual cages to remove group effects. All experimental procedures were in compliance with Guidelines for the Use of Animals in Neuroscience Research, 1992. All efforts were made to minimize the number of animals used and their suffering. Behavioral tests were started 3 weeks after AAV administration.

**Restraint stress model in rats.** Animals were housed singly in a natural light–dark cycle (12 h light and 12 h dark). The body weight of each rat was recorded every 7 days. Restraint stress was administered as described before[29,68,69]. Briefly, rats were immobilized for 6 h each day for 3 weeks during the light phase of the light:dark cycle. Rats were transported out of the colony room to an adjacent, brightly lit, quiet room and were placed into restraining tubes. The restrainers were transparent Plexiglas tubes measuring 24 cm long and 6 cm diameter with closing flaps and air holes. Corticosterone levels in serum samples were quantified with enzyme-linked immunosorbent assay using a commercial kit for corticosterone (Enzo Life Sciences). Rats were tested with the FST as previously described[70]. All efforts were undertaken to minimize possible suffering of animals.

**Mouse model of stress-induced anhedonia.** Studies were performed using 3-month-old male C57BL/6J mice. Three-month-old male CD1 mice were used as resident intruders for social stress, and 2–5-month-old Wistar rats were used for predator stress. C57BL/6J mice were housed individually under a reversed 12-h light–dark cycle (lights on: 21:00 hours) with food and water ad libitum, under controlled laboratory conditions ($22 \pm 1$ °C, 55% humidity). Experiments were carried out in accordance with the European Communities Council Directive for the care and use of laboratory animals and approved by the respective local governmental bodies (permission 0421/000/000/2013 was issued by General Directory of Ethical Committee of the New University of Lisbon, in accordance with Portuguese Law-Decrees DL129/92 (July 6), DL197/96 (October 16), and Ordinance Port.131/97 (November 7). All efforts were undertaken to minimize possible suffering of animals.

**Chronic stress procedure.** We employed a 10-day stress protocol[71] comprising a dark-cycle rat exposure stress (predator stress) and light-cycle semi-random application of two of three stressors: social defeat stress, restraint stress, and tail suspension stress. Animals were carefully monitored for physical state and body weight twice a day during the course of the study. Briefly, between the hours of 09:00 and 18:00 two stressors per day were employed in the following sequence: social defeat for 30 min, restraint stress for 2 h, and tail suspension for 20 min with an inter-session interval of at least 4 h (see below). All mice were carefully observed after each stress session and each morning and evening and weighed. Twelve hours after the last stressor, all mice were tested in 8-h sucrose test to assess hedonic traits. Immediately thereafter, their forced swimming was assessed to evaluate changes in affective behavior.

**Sucrose preference test.** Mice were given 8 h of free choice between two bottles of either 1% sucrose or standard drinking water, both 3 days prior the start of the stress study and at the end of day 10 of the stress paradigm. Other conditions of the test were applied as described elsewhere[71]. Percentage preference for sucrose is calculated using the following formula: Sucrose preference = volume (sucrose solution)/(volume (sucrose solution) + volume (water)) × 100.

**Forced swim test.** The Porsolt FST has been modified to prevent behavioral artifacts caused by stress-induced hyperlocomotion[72]. Floating behavior was defined as the absence of directed movements of the animal's head and body and was measured by visual observation that was validated previously by automated scoring with the CleverSys software (CleverSys, VA, US).

**Open field test.** Open field test for mice was performed as previously described[73].

**Stereotactic injection in the PFC of mice and 8-OH-DPAT treatment.** AAVs at the final concentration of 109 viral genomes per one μl were administered intracortically using TSE stereotaxic instrument (TSE Systems, Germany) according to the following coordinates: AP: +1.5 mm, L: +1.0 mm, DV: 2 mm and AP: +1.5 mm, L: +1.0 mm, DV: 2 mm. Prior operation, mice were anesthetized with 400 μl of 2.5% solution of the 2,2,2-tribromethanol (Sigma-Aldrich, T48402) and 2-methyl-2-butanol (Sigma-Aldrich, A1685) mixture (1:1). The volume of intracortically administered liquid was 2.0 μl in each hemisphere. Mice of control groups received 2.0 μl of pure vehicle or scrambled shRNA-based AAVs in each hemisphere. In several experiment, FST was performed in control animals or in mice injected with the AAVs after acute treatment with 5-HT1AR agonist 8-OH-DPAT (1 mg/kg, intraperitoneally) or saline performed 20 min before the test.

**Cell culturing and transfection.** Mouse neuroblastoma clone N1E-115 (N1E cells) (ATCC® Number: CRL-2263TM) were grown in the complete Dulbecco's modified Eagle's medium (DMEM; Invitrogen) supplemented with 10% fetal bovine serum (Gibco) and 1% penicillin/streptomycin (Gibco) at 37 °C and 5% $CO_2$. The cells were transfected using Lipofectamine™2000 Reagent (Invitrogen, Carlsbad, CA) according to the manufacturer's protocol. Primary hippocampal neuronal cultures were prepared as previously described[74] and transfected using Lipofectamine™2000 according to the manufacturer's manual.

**List of antibodies.** The following antibodies were used: 5-HT1AR (1:200 western blot (WB), 1:200 immunocytochemistry (ICC), 1 μg/ml IP; Alomone Labs (ASR-021), 5HT1A receptor (1:200 WB; Abcam ab64994), p44/42 MAPK (Erk1/2) (1:1000 WB; Cell Signaling Technology 9102), phospho-p44/42 MAPK (Erk1/2) (Thr202/Tyr204) (1:1000 WB; Cell Signaling Technology 9101), GFP (1 μg/ml for IP; GeneTex GTX26556), GFP (Horseradish peroxidase (HRP)) 1:1000 WB; GeneTex LS-C50850–500), HA probe (Y-11) Santa Cruz sc-805 (2 μg IP), synaptophysin 1 (1:50 ICC; Synaptic Systems 101 002), biotin (HRP; 1:500 WB; Sigma A4541), TGN38 (1:100 ICC; Thermo Scientific MA3–063), HA-peroxidase, high affinity (3F10) (1:8000 WB; Roche 12 013 819 001), Alexa Fluor® 647 mouse anti-GM130 (1:100 ICC; BD Biosciences 558712), ZDHHC9 polyclonal antibody (1:200 WB; Thermo Fisher Scientific PA5–26721), ZDHHC21 polyclonal antibody (1:200 WB; Thermo Fisher Scientific), and DHHC5 antibody (1:200 WB; ProSci Incorporated 54–211). NCAM antibodies[75] were kind gifts from Dr. Martina Muehlenhoff.

**DNA constructs, AAV vectors, and intracortical administration.** The 23 DHHC proteins were a kind gift from Masaki Fukata[76]. Plasmids encoding for shRNA for specific knockdown of different *Zdhhcs* were designed by Oligoengene (Seattle, WA, USA). Double-stranded oligonucleotides containing a target sequence were inserted into the pSUPER.neo+GFP vector for scrambling into the pSUPERIOR.retro.neo+gfp. The target sequences for selected shRNA constructs were as follows: Scrambled—5'-GCGCGCTTTGTAGGATTCG-3'; DHHC5—5'-ACCTCCACCTCCTATAAGA-3'; DHHC9—5'-GAGGAAAGTGGAAGTCGAC-3'; NS DHHC21—5'-CAAGATCCCACATGCAGAG-3'. To generate knockdown of the Zdhhc21 gene in vivo in mouse brains, mouse-specific shRNA 5'-CAAGATCCCACATGCAGAG-3' was subcloned into the pAAV-Syn(0.5)-EGFP-H1-2 vector[14] at BglII and SalI restriction sites. Similarly, scrambled shRNA 5'-ACTACCGTTGTTATAGGTG-3' was subcloned to be used as a control. The resulted plasmids were co-transfected with DJ vector and pHelper (Cell Biolabs Inc, USA) in HEK293-FT cells (ATCC) using Lipofectamine-3000 (Thermo Fisher Scientific Inc.). AAVs were collected after 48 h and then used for local intracortical infection of C57BL/6J mice in vivo. AAVs were administered intracortically using TSE stereotaxic instrument (TSE Systems, Germany) according to the following coordinates: AP, 1.5 mm and L, −1.0 mm; DV, 1 mm and AP, 1.5 mm; L, +1.0 mm and DV, 1 mm[77]. The volume of intracortically administered liquid was 2.5 μl in each hemisphere. Mice of control groups received 2.5 μl of pure vehicle or scrambled shRNA-based AAVs in each hemisphere. To confirm AAV infection, acute brain slices were subjected to microscopic analysis. Before analysis, animals were perfused with paraformaldehyde solution.

**Metabolic labeling and IP.** N1E cells were co-transfected with 5-HT1AR-CFP and plasmids encoding for individual DHHC-HA enzymes. Twenty-four hours after transfection, cells were labeled in serum-free DMEM containing [³H] palmitic acid (300 μCi/ml, 30–60 Ci/mmol) (American Radiolabeled Chemicals Inc.) for 2 h at 37 °C and 5% $CO_2$. Cells were lysed in RIPA buffer (pH 7.4), and 5-HT1AR was immunoprecipitated. Receptor was eluted, and samples were subjected to sodium dodecyl sulfate (SDS)-polyacrylamide gel electrophoresis and autoradiography.

**Erk1/2 phosphorylation assay.** Twenty-four hours after transfection, N1E cells were stimulated with 10 μM serotonin (5-HT) (Sigma) for 5 min. Cells were lysed in ice-cold lysis buffer containing 150 mM NaCl, 50 mM Tris/Cl, 0.5% sodium deoxycholate, 1% Triton X-100, 0.1% SDS, 1 mM sodium orthovanadate, 50 mM NaF, and 10 mM di-monosodium phosphate (pH 7.5) with freshly added PIs and 1 mM PMSF. Immunoblots were analyzed using p44/42 MAPK (Erk1/2) and phospho-p44/42 MAPK (Erk1/2) (Thr202/Tyr204) antibodies.

**Analysis of protein palmitoylation by ABE assay**. To analyze palmitoylated proteins in recombinant system as well as in native tissue, the ABE assay (Supplementary Fig. 1) was prepared as previously described[14].

**Analysis of cAMP response**. N1E cells expressing the mCerulean-Epac-Citrine (CEPAC) biosensor[18], 5-HT1AR-mCherry, and shRNAs against individual ZDHHCs tagged with GFP were used for live cell imaging with the Zeiss LSM780. After 2 min of acquisition, cells were pre-stimulated with 1 μM forskolin (Sigma) and 25 μM 3-isobutyl-1-methylxanthine (Sigma) to increase intracellular cAMP to sub-saturation levels. After 6 min, when cAMP reached a constant level, 5-HT1AR was stimulated with 20 μM 5-HT (Sigma), and the change in acceptor/donor fluorescence intensity ratio of CEPAC was obtained in online fingerprinting mode and further processed as previously described[78].

**Analysis of the intracellular distribution of 5-HT1AR and ZDHHCs**. N1E-115 cells were co-transfected with the GFP-tagged ZDHHC and 5-HT1AR-mCherry. Sixteen hours after transfection, cells were imaged using LSM780. Offline analysis of fluorescence intensity distribution was carried out using the Fiji software. To quantify fluorescence intestines of GFP and mCherry at the Golgi compartment and at the plasma membrane, two ROIs (regions of interest) corresponding to these organelles were drawn. The median intensity of each channel for each ROI was quantified and the ratio of membrane versus Golgi intensity was calculated separately for each protein.

**Patch-clamp recordings from hippocampal neurons**. On day in vitro 8, primary hippocampal neurons were transfected with either a control pSUPER.neo+GFP/scrambled shRNA plasmid (1 μg per well) or with shRNA against ZDHHC21 (1 μg per well, pSUPER expression vector) to silence the expression of endogenously expressed ZDHHC21. Neurons were used for electrophysiological recordings 3–4 days after the transfection. Whole-cell recordings from pyramidal neurons were obtained using electrodes with a resistance of 3–4 MOhm. The electrodes were filled with a solution comprising the following components (in mM): 130 K-gluconate, 8 NaCl, 4 Mg-ATP, 0.3 Na-GTP, 0.5 EGTA, 10 HEPES, pH 7.25. Cells were placed in HEPES-buffered saline with the following composition (in mM): 119 NaCl, 5 KCl, 2 CaCl₂, 2 MgCl₂, 25 HEPES, 20 D-glucose, 0.0005 Na⁺ channel blocker tetrodotoxin citrate (Tocris), 0.001 5-HT7R antagonist SB-269970 (Tocris), pH 7.3. Currents were recorded with an EPC 10 USB Patch Clamp Amplifier (HEKA Elektronik). Data acquisition was controlled using the PATCHMASTER software (HEKA Elektronik). The mean amplitudes of currents of three independent experiments activated 180–200 ms after the beginning of the voltage step were measured and statistically evaluated using Student's $t$ test.

**Palmitoylomics**. Proteins purified from ABE approach were dissolved in 2% SDS Buffer, diluted with 19 volumes of 50 mM TRIS, 100 mM NaCl, and 1 mM EDTA, and digested in solution with trypsin (GOLD TRYPSIN for MS) by incubating at 37 °C for 60 min. The resulting peptide mixtures were applied to RP-18 pre-column (Waters, Milford, MA) using water containing 0.1% formic acid (FA) as a mobile phase and then transferred to a nano-high-performance liquid chromatography (LC) RP-18 column (internal diameter 75 μM, Waters, Milford, MA) using acetonitrile (ACN) gradient (0–35% ACN in 160 min) in the presence of 0.1% FA at a flow rate of 250 nl/min. The column outlet was coupled directly to the ion source of Q-exactive Orbitrap mass spectrometer (Thermo Electron Corp., San Jose, CA) working in the regime of data-dependent MS to MS/MS switch. A blank run ensuring absence of cross-contamination from previous samples preceded each analysis.

All MS data sets were searched against SwissProt protein database (Swissprot 2018_02; 16,905 sequences) using the MASCOT search engine (MatrixScience, London, UK, Mascot Server 2.5.1). Both the peptide and fragment mass tolerance settings were established separately for individual LC-MS/MS runs after a measured mass recalibration, as described previously[79]. After recalibration, the mass tolerance for proteins was in the range 5–10 ppm and for peptides 0.01–0.05 Da. The Mascot search parameters were as follows: enzyme, Trypsin; missed cleavages, 1; variable modifications N-Ethylmaleimide (C), carbamidomethyl (C), Oxidation (M); instrument, HCD; Decoy option, active. False discovery rate (FDR) was estimated with Mascot Decoy search and score threshold was adjusted for each sample to keep the FDR <1%. Probable contaminants (e.g., keratins and albumin) were removed from the identified protein/peptide lists.

The MS data have been deposited to the ProteomeXchange Consortium via the PRIDE partner repository (http://www.ebi.ac.uk/pride/archive/) with the data set identifier PXD012736. For evaluation of the relative protein abundance in each sample, spectral count values determined using emPAI scores were used as previously described[31,80]. For data sets generated by the protein-based procedure, spectral counts were merged over all biological replicates. Subsequently, statistical analysis was performed on the merged data sets. For the protein data set, missing values were imputed; all zeros were replaced with 1. Only proteins that met the acceptance criteria: FDR < 1%, at least two unique peptides, Mascot score >25, non-redundant proteins, were taken for further analysis. The protein-wise natural logarithms (ln) of specific ratios were calculated. Significance were computed ($P$ values) using variance $t$ test. Proteins with $P$ values <0.05 were accepted as S-palmitoylated candidates (Supplementary Data 1 and 2). The statistical significance

of the difference between control/scramble/shDHHC21 (enriched, reduced, no influence) was calculated using the unequal variance two-tailed heteroscedastic $t$ test. It was assumed that $\log_2$ shDHHC21/scramble and/or $\log_2$ shDHHC21/control and or scramble/control <−2.0 (5% threshold) and −1.0 (105 threshold) reflected reduction when >1.0 and 2.0 reflected induction of palmitoylation of the given protein. Then statistical significance of the difference between spectral counts for each protein was estimated with the unequal variance one-tailed, heteroscedastic $t$ test. It was assumed that a palmitoylated protein was identified with high confidence, when the $P$ value of the difference between two biological conditions was ≤0.05 or ≤0.001 (Supplementary Data 2). All palmitoylomics raw data are deposited at the PRIDE repository (PXD012736).

**Quantitative real-time PCR analysis**. Total RNA was isolated using RNeasy® Plus Mini Kit (QIAGEN) according to the manufacturer's protocol. RNA was reverse transcribed into cDNA using SuperScript® III First-Strand Synthesis System (Invitrogen), and the expression analysis was performed using TaqMan® Gene Expression Assay (Applied Biosystems). For the detection of Zdhhc5, Zdhhc9, and Zdhhc21, the gene-specific primers were Mm00523158_m1, Mm00552609_m1, Mm00509795_m1, Rn01747138_m1, Rn01462327_m1, and Rn01764799_m1 (Applied Biosystems), respectively. The following primers were used for detection of ZDHHC in human samples: ZDHHC5, Hs00379814_m1; ZDHHC9, Hs00211318_m1; and ZDHHC21, Hs00944036_m1. The relative gene expression was calculated using the ΔΔCT method.

**miRNA analysis**. To establish the importance of the predicted miRNAs (see Supplementary Table 1) in regulating the Zdhhc21 expression, we mimicked a miRNA environment in N1E cells by overexpressing the miRNAs of interest. N1E cells were transfected with specific pre-miR™ miRNA Precursor Molecules (Ambion, Applied Biosystems) by a liposomal-based method (Lipofectamine™2000, Invitrogen) according to the manufacturer's instructions. After 72 h, cells were washed and RNA was isolated and subjected to RT-PCR analysis.

For analysis of miRNA expression in human postmortem samples, total RNA from the mPFC was isolated using the miRNeasy Mini Kit (Qiagen) according to the manufacturer's instructions. Quantification of RNA was performed with Synergy HT Reader (BioTek). Single miRNA expression analysis was validated using TaqMan MiRNA assays (Applied Biosystems) for miR-30a, miR-30e, and miR-200a in a two-step approach. Real-time PCR was performed on a QuantStudio system (Applied Biosystems). For internal normalization of miRNA expression levels, housekeeping RNA RNU48 was chosen, and relative quantification was performed using internal dilution series for each single target.

**Reporting summary**. Further information on research design is available in the Nature Research Reporting Summary linked to this article.

## Data availability
The data that support the findings of this study are available from the corresponding authors upon reasonable request. Palmitoylomics raw data are stored on the PRIDE repository http://www.ebi.ac.uk/pride/archive/projects/PXD012736 (PXD012736). The source data underlying Figs. 1, 2, 3, 4, 5, 6 are available as a Source Data file.

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

## Acknowledgements

We thank Dr. Nicholas Davis for his help in establishing the ABE assay and Dr. Masaki Fukata for providing ZDHHC clones. We thank Elena Zakharova and Viktoria Ravkina for technical support and Dr. Christo Goridis and Dr. Martina Muehlenhoff for providing anti-NCAM antibodies. This study was supported by the Deutsche Forschungsgemeinschaft (DFG) grant PO732 to E.P.; by the Basic Research project #0324–2019–0041 and Russian Scientific Foundation (grant No 19-15-00025) to V.S.N.; and by "5–100" Russian Academic Excellence Project, NARSAD and the Brain and Behavior Research Fund, Young Investigator Award to T.S. T.T. was supported by a Consolidator ERC grant from the European Union, and T.T. and E.P. were supported by the REBIRTH Excellence cluster. J.W., A.K., and M.Z. acknowledge the National Science Centre grants: UMO-2017/26/E/NZ4/00637. M.B. has been partially supported by the Polish Ministry of Science (1342/1/MOB/IV/15/2016/0).

## Author contributions

E.P., G.P., D.R., S.S.: experimental concept, manuscript writing; E.P.: team coordination, data analysis; N.G., D.A.G., M.B.: biochemical and ABE analysis, cell imaging; G.P.: human sample analysis; S.P., A.Z.: biophysical analysis; A.W.: electrophysiological experiments; T.T., C.B., L.H., J.F.: miRNA analysis; A.D., G.K., F.P., D.S.: fear conditioning experiments; I.S.: experiments with ZDHHC21-deficient mice; T.S.: depression model mice experiments; M.B., J.W., A.K.: experiments in rats and qPCR analysis; M.Z.-K., J.W.: palmitoylomics; V.S.N., D.B., T.I., E. Kondaurova, E. Kulikova: experiments in mice and qPCR analysis.
