## [Peer Review File with Redaction · Nature Communications]

Editorial Note: Parts of this Peer Review File have been redacted as indicated to maintain patient confidentiality.

Reviewers' comments:

Reviewer #1 (Remarks to the Author):

General comments

The authors focused on elucidating the molecular mechanism and clinical relevance of 5HT1A receptor palmitoylation in prefrontal cortex. The 5HT1A receptor is well validated drug target (buspirone is a clinically used drug and 5HT1A partial agonist). Prefrontal cortex is involved in feedback response to stress and PFC changes are indeed observed in rodent and human PET/MRI studies. Therefore the research presented in this manuscript is timely and of importance. The manuscript is generally well written, the flow of the experiments is logical, and the molecular axis ZDHHC21 -> 5HT1A palmitoylation is well validated in vivo and in vitro. Correlation between ZDHHC21 -> 5HT1A palmitoylation down-regulation and depressive like symptoms is well elaborated. The authors went further in exploring the mechanism by detecting the miR30e epigenetic regulation in vitro. Propagation to the post mortem human studies may be of clinical interest in the future as well. There are however several issues that have to be considered, including additional experiments, by the authors as detailed below.

Specific comments

1. The causality between ZDHHC21 -> 5HT1A palmitoylation down-regulation and depressive like symptoms in mice is not based on solid evidence. The authors used series of behavioral paradigms including sucrose preference, CORT levels etc. However, ZDHHC21 PFC knockdown was not phenotyped in detail. Why? More extensive phenotyping of this animal is needed to conclude that it shows depressive-like phenotype since forced swim test alone is an antidepressant screening test and depressive-like phenotype cannot be claimed based on this test alone. Next, Figure 5C represents a picture and it doesn't offer quantitative explanation to the statement "5-HT1A palmitoylation was selectively reduced in the PFC" (please provide bargraph + stat at least in addition). Next, the reviewer is not convinced that the expression of 5HT1A in the PFC is unchanged when looking at the figure S7G, variability is huge (error bars represent SEM) and the authors do not have sufficient statistical power to claim equivalency here.
2. Druggability of ZDHHC21 -> 5HT1A palmitoylation is questionable. Buspirone and 8OH DPAT are 5HT1A agonists which are known to reverse depressive like phenotypes (FST in particular). Would the authors be able to do so using the ZDHHC21 PFC knockdown? Can overexpression of ZDHHC21 in the PFC (viral etc) in rodents offer resilience to stress/depressive-like phenotype? In order to claim that ZDHHC21 -> 5HT1A palmitoylation is a valid drug target, the authors have to answer these questions
3. miR30e epigenetic regulation is not shown in vivo. Is miR30e overexpressed in the PFC of the rodent depression models?
4. More information about the suicide and control samples are necessary. For which parameters were these samples matched (age, gender, BMI etc)?
5. In mice all three Zdhhc21 (5, 9 and 21) are affected in anhedonic mice and rats with depression like symptoms, suggesting the possibility of functional redundancy, particularly in light of the results presented in Figure 1. However, the authors clearly prefer Zdhhc21 throughout the manuscript and only this factor is followed up on with functional tests. The authors should further substantiate the reasons for this focus. Also, these data incentivize the use of triple knock-downs against all three Zdhhc21 to potentially amplify phenotypic manifestations.
6. The authors show that Zdhhc21 knock-down in the in PFC results in the development of depressive symptoms. It would strongly elevate the study if the authors could demonstrate that enhancing Zdhhc21 levels can rescue depression in murine models. To this end, they could use e.g. a similar experimental setup but injecting antagomirs against miR-30e, which the authors claim targets Zdhhc21.
7. The functional effect of the depilated mouse (ref. 17) is hypothesized to be due to protein mislocalization. The authors should check intracellular localization of Zdhhc21 in dep/dep mice in the brain. Also they should

subject these dep/dep mice to their models of depression-like symptoms.

8. While tuning of expression levels by miRNAs is indeed occasionally referred to as an “epigenetic mechanism”, the reviewer strongly suggest to use a different more accurate term, such as “transcriptional regulation”. See e.g. PMID 29339796. <https://www.nature.com/articles/nrm.2017.135.pdf>

9. In Figure 1B it is suggested to also show effects of the other DHHCs beyond 5, 9 and 21 (i.e. Figure S2B). Why is #9 considered positive but #7 is not?

10. Is the gel image in Figure 1E from a single blot or is it stitched together? There appear to be breaks e.g. between lanes 2 and 3 and between lanes 4 and 5 as well as 5 and 6 in the upper Palm 5-HT1AR row.

Indeed one gets the impression that the –lanes have been obtained by covering the corresponding band or stitching different pieces together. This has indeed to be clarified by the authors showing the entire gel in e.g. supplementary documentation.

11. On page 5 it is stated “Noteworthy, knock-down of ZDHHC21 gave the more prominent reduction of 5-HT1AR palmitoylation, confirming this ZDHHC as a major palmitoyl-transferase for 5-HT1AR.” From Figure 1E it appears that DHH9 knock-down seems to result in the strongest effect.

Reviewer #2 (Remarks to the Author):

The manuscript by Gorinski et al showed that serotonin 1A receptor (5-HT1AR) is palmitoylated in the brain and its palmitoylation level is specifically reduced in the prefrontal cortex (PFC) in rodent models of depression and in the PFC of depressed suicide subjects. The authors identified ZDHHC21 as a major palmitoyl-transferase for 5-HT1AR and found the reduced expression of ZDHHC21 in the above rodent models and suicide subjects. Then, the authors demonstrated the causative relationship among reduced ZDHHC21 expression in the PFC, 5-HT1AR palmitoylation level, and depressive symptoms. This work proposes an interesting, new molecular mechanism for the pathogenesis of depressive symptoms. Addressing the following points would strengthen this paper.

Major comments

(1) The authors showed that knockdown of ZDHHC21 in PFC reduced the 5-HT1AR palmitoylation (Fig. 5C) and that a global protein palmitoylation was not affected by ZDHHC21 knockdown (Supplementary Fig. 7J). However, the silver staining presented in Supplementary Fig. 7J is not sufficient to neglect the involvement of palmitoylated proteins other than 5-HT1AR. Quantitative palmitome analysis combined with mass spectrometry is useful to confirm the specificity and their claim.

(2) Knockdown of ZDHHC21 may have target off effects. Does loss of function mutant mouse of ZDHHC21 (used in Fig. 1F) display the similar depression-like behavior?

(3) The authors proposed that microRNA (miR-30e) might be involved in the regulation of ZDHHC21 expression as a negative regulator. To prove this hypothesis, the authors should examine whether miR-30e is increased in the rodent models of depression.

Minor comments

(1) In the abstract, the second sentence “Here we demonstrated that 5-HT1AR is palmitoylation in human and rodent brains” should be “Here we demonstrated that 5-HT1AR is palmitoylated in human and rodent brains”.

(2) On page 7, line 7 from the bottom, “Supplementary Fig. 6” should be “Supplementary Fig. 5E”. As one of general readers, I suggest that the description/citation of Supplementary Figure should be specifically indicated throughout the manuscript, such as “Supplementary Fig. 5E”, but not “Supplementary Fig. 5”.

(3) In the supplementary figures, on page 9, line 3 from the bottom, (E) is (J).

Reviewer #3 (Remarks to the Author):

This study examines palmitoyl-transferase ZDHHC21 in 5-HT1A palmitoylation and depression phenotypes. Depletion of ZDHHC21 reduces 5-HT1A palmitoylation and signaling, and both are reduced in depressed PFC. Knock-down of ZDHHC21 in PFC results in depression-related behavior. miR30e is identified as a negative regulator of ZDHHC21.

The authors present extensive data supporting the effects of ZDHHC9 and 21 in 5-HT1A palmitoylation, and a partial inhibition of 5-HT1A signaling in transfected cells, with a striking effect on 5-HT1A-GIRK coupling in hippocampal neurons. Overall their data support the argument that reduced ZDHHC21 in PFC could mediate a depression phenotype, but the exact role of 5-HT1A palmitoylation is unclear. Furthermore the role of miRNA in stress-induced alterations is unclear.

Since ZDHHC21 may have many palmitoylation targets (e.g., see Kang R et al., Nature 456: 904, 2008), to show that 5-HT1A palmitoylation is critical for the depression phenotype following its depletion, studies need to be done on a 5-HT1A $-/-$ background to block the effect of ZDHHC21 depletion. Alternately, the effect of knock-in or rescue of a palmitoylation-defective 5-HT1A mutant could be tested.

Several minor grammatical errors need correction: e.g., Abstract, use present tense: Here we demonstrate that 5-HT1A is palmitoylated... and identify ZDHHC21...; avoid run-on sentences (e.g. the above sentence could be split).

Specific comments:

1. Introduction: it is important to note that over-expression of 5-HT1A in Gross et al. was done in pyramidal cells and in early post-natal mice.
2. Results, p. 4: in the screen of PTs, ZDHHC5, 9, and 21 enhanced 5-HT1A palmitoylation: are these PTs conserved, do they form a sub-family?
3. Supp. Fig. 2B: What are the #; show error bars; what was the N-value? ZDHHC5 seems to have the greatest effect on 5-HT1A palmitoylation, was this consistent or significant? Some of the other ZDHHCs seem to have activity, why were they excluded? What was the rationale for focusing on ZDHHC21?
4. Supp. Fig. 2D: ZDHHC9/21 appear to relocalize 5-HT1A to Golgi, while ZDHHC5 appears to maintain its localization to PM. Quantify.
5. Results, p. 5: In terms of signaling, G α -i subunits are palmitoylated, which is crucial for receptor-G protein signaling (e.g., see Schattauer SS, Nature Comm 2017); did depletion of ZDHHCs affect G α palmitoylation?
6. Fig. 1: in fig. 1b there seems to be endogenous palmitoylation of 5-HT1A, yet in 1d no palmitoylation was detected in the "mix" samples: explain. What post-test was used?
7. Fig. 2C: define relative cAMP amplitude: relative to pretreatment?
8. Fig. 3, 4: show timeline for stressors and provide more details for the sequence and frequency of stressors, especially in mice. Quantify levels of Total 5-HT1A receptors: was the reduction in palmitoylated 5-HT1A simply reflecting a reduction in total 5-HT1A? What is the evidence of the specificity of the 5-HT1A antibody for 5-HT1A (e.g., is there any 5-HT1A staining in 5-HT1A knockout mice)?
9. Fig. 5B: How large was the spread of infection: did it cover the entire PFC, were other cortical or hippocampal areas infected? The protocol needs to be described in detail.
10. Fig. 6: In the postmortem samples, was the expression of 5-HT1A differing in control vs. suicide PFC?
11. Figs. 3, 4, 6: Is miR30e or 30a altered in depression? What is the effect of blocking miR30e on stress induced depression?
12. Discussion: It is important to emphasize that studies showing depression associated with increased 5-HT1A were in the raphe (autoreceptors). Note one study that did show impaired 5-HT1A signaling in depressed suicides should be mentioned (Hsiung SC et al., J. Neurochem. 2003).
13. Discussion: The data in Fig. 3 indicates that a 5-HT1A receptor lacking palmitoylation sites can still couple to cAMP or ERK1/2, hence palmitoylation is not essential for coupling but enhances coupling, and does not down-regulate coupling, but partially attenuates it.
14. Discussion: The data do not resolve whether the effect of reduced ZDHHC21 on depression is mediated solely by reduced 5-HT1A palmitoylation.

Reviewers' comments:

Reviewer #1 (Remarks to the Author):

General comments

The authors focused on elucidating the molecular mechanism and clinical relevance of 5HT1A receptor palmitoylation in prefrontal cortex. The 5HT1A receptor is well validated drug target (buspirone is a clinically used drug and 5HT1A partial agonist). Prefrontal cortex is involved in feedback response to stress and PFC changes are indeed observed in rodent and human PET/MRI studies. Therefore the research presented in this manuscript is timely and of importance. The manuscript is generally well written, the flow of the experiments is logical, and the molecular axis ZDHHC21 -> 5HT1A palmitoylation is well validated in vivo and in vitro. Correlation between ZDHHC21 -> 5HT1A palmitoylation down-regulation and depressive like symptoms is well elaborated. The authors went further in exploring the mechanism by detecting the miR30e epigenetic regulation in vitro. Propagation to the post mortem human studies may be of clinical interest in the future as well. There are however several issues that have to be considered, including additional experiments, by the authors as detailed below.

Specific comments

1. The causality between ZDHHC21 -> 5HT1A palmitoylation down-regulation and depressive like symptoms in mice is not based on solid evidence. The authors used series of behavioral paradigms including sucrose preference, CORT levels etc. However, ZDHHC21 PFC knockdown was not phenotyped in detail. Why? More extensive phenotyping of this animal is needed to conclude that it shows depressive-like phenotype since forced swim test alone is an antidepressant screening test and depressive-like phenotype cannot be claimed based on this test alone.

The Reviewer makes a valuable suggestion here. According to that, we greatly extended our behavioral characterization of animals after ZDHHC21 knockdown. To that end, we analyzed three cohorts: (i) animals injected in PFC with either vehicle, or (ii) AAV constructs encoding for scramble shRNA or (iii) shRNA against ZDHHC21. In addition to forced swimming test (FST) mentioned in initial manuscript, we performed the following behavioral experiments: open field test to separately analyze (1) the general locomotor activity, (2) exploration and (3) the anxiety-like behavior, (4) tail suspension test, and (5) the novel object recognition test. Supporting our initial observation, only in group with knocked-down ZDHHC21, we obtained an augmentation of the depression-related behavior as assessed by the FST. It is also noteworthy that in all other behavioral tests we did not observe any significant differences between the groups. These data are mentioned on page 10 of the revised manuscript and shown in Supplementary Figure 7J to M.

Next, Figure 5C represents a picture and it doesn't offer quantitative explanation to the statement "5-HT1A palmitoylation was selectively reduced in the PFC" (please provide bargraph + stat at least in addition).

We have quantified these data as requested and included quantification in the revised Figure 5D. Indeed, these data show significant (vehicle vs. shDHHC21 $P = 0.02$; scramble vs. shDHHC21 $P = 0.01$) decrease of 5-HT1AR palmitoylation in the PFC.

Next, the reviewer is not convinced that the expression of 5HT1A in the PFC is unchanged when looking at the figure S7G, variability is huge (error bars represent SEM) and the authors do not have sufficient statistical power to claim equivalency here.

To increase statistical power, we combine our initial data with the results of two additional experiments, each of which included at least 9 animals injected with AAV constructs encoding

either for vehicle, or scramble shRNA or shRNA against ZDHHC21. After behavioral analysis mentioned above, the PFC lysates from experimental animals were analyzed by the Western blot to quantify expression of the 5-HT1A protein (buffer vs. shDHHC21, $p=0.37$; buffer vs. scr, $p=0.55$; scr vs. shDHHC21, $p=0.09$. One way ANOVA). Thus, based on these data, which are summarized in the revised Figure 5E, we now have a clear statistic basis to state that there are no significant differences of 5-HT1A expression.

2. Druggability of ZDHHC21 -> 5HT1A palmitoylation is questionable. Buspirone and 8OH DPAT are 5HT1A agonists which are known to reverse depressive like phenotypes (FST in particular). Would the authors be able to do so using the ZDHHC21 PFC knockdown?

The main reason why we didn't address the druggability of ZDHHC21-mediated 5-HT1A palmitoylation in initial study was mainly due to our previous observation that palmitoylation of the 5-HT1A is not modulated by the receptor stimulation with agonists (Papoucheva et al. (2004) J Biol Chem. 279(5):3280-91). This suggests that the anti-depressive action of buspirone and 8-OH-DPAT is not mediated via modulation of 5-HT1A palmitoylation.

We, however, followed Reviewer's suggestion and analyzed the acute effect of 8-OH-DPAT on behavior of ZHHC21 PFC knockdown in order to verify substrate specificity of ZDHHC21 towards the 5-HT1A. As mentioned by Reviewer, multiple prior studies evidently show that acute 8-OH-DPAT injection results in a 5-HT1A-mediated anti-depressive effects as assessed by decreased immobility time in the FCT (e.g. Borsini (1995) Neurosci Biobehav Rev; O'Neill and Conway (2001) Neuropsychopharmacology; Sugimoto et al (2010) Eur J Pharmacol; Miyake et al (2014) Pharmacol Biochem Behav). We were able to reproduce this 5-HT1A-mediated effect of 8-OH-DPAT in control conditions, when DHHC21 was normally expressed. However, acute injection of 8-OH-DPAT failed to induce any anti-depressive effects in animals after knock down of DHHC21 (which results in significant reduction of 5-HT1A palmitoylation in the PFC and, consequently, in impaired receptor functions). This data provides a direct evidence for ZDHHC21-mediated 5-HT1A palmitoylation playing a central role in the etiology of depressive phenotype. In the revised manuscript, we included 8-OH-DPAT data into the "Results" section on page 12 and presented these in supplementary figure 7N. In addition, these results are discussed on page 16.

3. miR30e epigenetic regulation is not shown in vivo. Is miR30e overexpressed in the PFC of the rodent depression models?

We thank the Reviewer for this suggestion. We now analyzed the expression profile of miR-30a, miR-30e and miR-200a. In these experiments we decided to use human post-mortem mPFC samples, in order to directly evaluate the role of these micro-RNAs for depression. These experiments revealed that the expression of miR30a and miR30e was significantly increased, while the expression of miR-200a was drastically reduced in samples from depressive suicide victim (Fig. R1 below). These new results provide a strong experimental support for our hypothesis on the role of defined micro-RNAs in development of depression, and we thank the reviewer for this important comment. The micro-RNA expression data are now shown in the revised Figure 6F and mentioned on pages 13, 16-17, and 26. In the future studies, we are planning to evaluate miR-30a, -30e, and -200a as potential biomarkers to identify MDD patients with suicidal thoughts.

Fig. R1. Analysis of miR-30a, -30e, and -200a expression in PFC of control and depressed suicide victims. (t-test, * $P < 0.05$, ** $P < 0.01$).

4. More information about the suicide and control samples are necessary. For which parameters were these samples matched (age, gender, BMI etc)?

It was due to ethical considerations that we decided to exclude a table containing detailed demographic information on the suicide victims and control persons from the main text of manuscript. Please find this data below in Table R1. In general, groups were matched for age (mean age value in control group was 44.2 ± 4.1 years, and in suicide group – 42.2 ± 4.3 years, with a similar age distribution within groups), gender (36% female in control and 43% in suicide group), and race (12.5% of African Americans in both groups). We now mentioned these parameters on page 12 of our revised manuscript. In case that the Reviewer is of a strong opinion that the demographic table will provide additional value for our manuscript, we would be prepared to add them to the supplementary data.

[REDACTED]

[REDACTED]

5. In mice all three *Zdhhcs* (5, 9 and 21) are affected in anhedonic mice and rats with depression like symptoms, suggesting the possibility of functional redundancy, particularly in light of the results presented in Figure 1. However, the authors clearly prefer *Zdhh21* throughout the manuscript and only this factor is followed up on with functional tests. The authors should further substantiate the reasons for this focus.

We apologize for not being sufficiently clear here. The rationality for our focus on ZDHHC21 was based on following observations:

(i) Our quantitative analysis shown in Figure 1C demonstrates that only over-expression of ZDHHC21 results in significantly increased 5-HT1AR palmitoylation.

(ii) When expression of ZDHHC5, 9 and 21 was evaluated in human samples obtained from depressive suicide victims as well as in both rodent depression models, we found that only expression of ZDHHC21 was significantly reduced throughout all the depression models. Indeed, in the human MDD samples, ZDHHC9 and 21 were affected, in the mouse model – only 21, and in the rat model ZDHHC5, 9 and 21.

We also quantified the effects of shRNAs against ZDHHC5, 9 and 21 on 5-HT1AR palmitoylation shown in Fig. 1E and F. In case of ZDHHC5 knock-down, we obtained a slight decrease in 5-HT1AR palmitoylation ($p = 0.049$). Knock-down of either ZDHHC9 or ZDHHC21 resulted in a more significant decreased receptor palmitoylation ($p=0.006$ for ZDHHC9 and $p=0.008$ for ZDHHC21 knock-down), suggesting that ZDHHC9 might also be involved in 5-HT1AR palmitoylation.

We hope that these new results in combination with above mentioned observations clarify our decision to focus on ZDHHC21 as an obviously more potent 5-HT1AR acyl-transferase. To clarify this point, we have now included the following statement to the results section:

“Noteworthy, knock-down of ZDHHC9 and -21 gave the more prominent reduction of 5-HT1AR palmitoylation in comparison to the effect mediated by shRNA against ZDHHC5 (Fig. 1F), suggesting that both ZDHHC9 and ZDHHC21 represent relevant palmitoyl-acyltransferases (PATs) for 5-HT1AR. Based on the results obtained after ZDHHC overexpression (Fig. 1B and C), we decided to focus on ZDHHC21 as a more potent PAT for the 5-HT1AR”.

Also, these data incentivize the use of triple knock-downs against all three Zdhhc3 to potentially amplify phenotypic manifestations.

When we analyzed efficiency and substrate specificity of ZDHHC21 towards 5-HT1AR by proteomics approach, we found that the knock-down of ZDHHC21 in the PFC of mouse resulted in a strong (app. 15-fold) and highly significant ($p=0.0028$) decrease of 5-HT1AR palmitoylation, demonstrating that ZDHHC21 is the more specific palmitoyl-transferase for 5-HT1AR in the mouse brain (Fig. R6 below). These results also suggest that the simultaneous knock-down of all three relevant ZDHHCs could provide only a very minor add-on effect, if at all. Moreover, triple knock-down might result in the off-target depalmitoylation, which could artificially influence behavioral response. Indeed, it has been shown that ZDHHC5 can palmitoylate a couple of neuronal proteins, including GRIP1 (Thomas et al., 2012, Neuron, 73), Delta-catenin (Brigidi et al., 2014, Nat. Neurosci. 17), and Flotillin-2 (Li et al., 2012, J. Biol. Chem. 287), while ZDHHC9 is the main acyltransferase for H- and N-Ras (Swarthout et al., 2005, J. Biol. Chem., 280).

6. The authors show that Zdh21 knock-down in the in PFC results in the development of depressive symptoms. It would strongly elevate the study if the authors could demonstrate that enhancing Zdh21 levels can rescue depression in murine models. To this end, they could use e.g. a similar experimental setup but injecting antagomirs against miR-30e, which the authors claim targets Zdh21. Can overexpression of ZDHHC21 in the PFC (viral etc) in rodents offer resilience to stress/depressive-like phenotype? In order to claim that ZDHHC21 -> 5HT1A palmitoylation is a valid drug target, the authors have to answer these questions *We fully agree with the Reviewer that a direct evidence for the possible therapeutic role of ZDHHC21 overexpression is an intriguing issue. Therefore, we recently overexpressed ZDHHC21 in the PFC of mouse subjected to the short-day conditions. In this depression model (which was successfully established in our lab), C57BL/6J mice developed a depressive-like behavioral phenotype after one month of housing with 4 (instead of 14) hour-long light period (Bazhenova et al. (2019) Neurosci. Letter). One group of mice was injected with a buffer (control), and another with AAV encoding for ZDHHC21. Thirty days after the injections, depression-like phenotype was evaluated using a forced swimming test (FST) followed by the 5-HT1AR palmitoylation analysis in the PFC. Results of these experiments (Fig. R2, see below) demonstrate that overexpression of ZDHHC21 prompts a significant increase of 5-HT1A palmitoylation in the PFC, which is accompanied by the significantly decreased immobility time in FST, while the general locomotor activity assessed by the open field test was not affected.*

These new data provide a first indication for ZDHHC21-mediated 5-HT1AR palmitoylation as a novel therapeutic target in the treatment of depression. Whilst we believe this quest requires a separate study, we might be able - if insisted upon - to include some of these data into the revised manuscript.

We, however, assume that the detailed characterization of therapeutic aspects will require more extensive and wide-ranging analysis, which would be out of scope of the present manuscript. Therefore, we plan to include results obtained with the short-day conditions depression model together with a detailed analysis of other depression models upon ZDHHC21 overexpression into follow-up manuscript.

Fig. R2. (A) Relative expression levels of *Zdhhc21* in PFC of depressive mice subjected to the short-day conditions (DEPR). (B) PFCs of depressive mice were isolated 30 days after injection either with vehicle (VEH) or AAV encoding for ZDHHC21 (OE DHHC21) and subjected to ABE assay to define 5-HT1AR palmitoylation. (C) Immobility time in the forced swim test for mice treated with vehicle ($n = 6$) or AAV encoding for ZDHHC21 ($n = 7$). Statistical significance between values is noted (* $P < 0.05$, two-tailed t-test). (D) Comparison of the motor activity assessed by the open field test in mice treated with vehicle ($n = 10$) or AAV encoding for ZDHHC21 ($n = 10$).

7. The functional effect of the depilated mouse (ref. 17) is hypothesized to be due to protein mislocalization. The authors should check intracellular localization of *Zdhhc21* in dep/dep mice in the brain. Also they should subject these dep/dep mice to their models of depression-like symptoms.

*This per se certainly is a justified suggestion. Since non-conditional transgenic models often show strong compensatory effects during development, we first analyzed whether the reduced 5-HT1AR palmitoylation obtained in the brains of newborn (P0) *Zdhhc21*^{dep/dep} animals persisted till adulthood. However, in contrast to the results obtained in P0 *Zdhhc21*^{dep/dep} animals, in the brains of adult (P30) mice no signs of any decrease of 5-HT1AR palmitoylation were observed (Fig. R3 below). Thus, our results demonstrate that Dep mice can hardly be used as an appropriated model to verify our hypothesis about the role of 5-HT1AR palmitoylation in depression. Also to note that no single publication out of the five dealing with the Dep mouse line mentioned any depression-like behavioral abnormalities. This was further confirmed during our personal communication with Dr. Ian Smyth and Dr. Ian Jackson, who demonstrated that the Dep mouse contains a mutation in the *Zdhhc21* gen (PLoS Genet. 2009, 5:e1000748). Importantly, these mice display a spectrum of pathological symptoms not related to brain functions, including disrupted epidermal homeostasis (PLoS Genet. 2009, 5:e1000748), reduced vascular tone, which manifests in vivo as hypotension and tachycardia (Arterioscler Thromb Vasc Biol. 2016, 36:370-9) and endothelial dysfunction (Nat Commun. 2016, 7:12823). These deficits would further complicate usage of *Zdhhc21*^{dep/dep} mice as an eligible depression model.*

In the revised manuscript, we added new results concerning 5-HT1AR palmitoylation in the brain of P30 Dep mouse to the Supplementary Figure 3E and discussed these data on page 6.

Fig. R3. Brain tissues isolated from adult (P30) $F233\Delta Zdhhc21^{dep/dep}$ ($n = 3$) and wild-type ($n = 3$) mice were collected for ABE analysis following by quantification. No significant differences in 5-HT1AR palmitoylation were obtained.

8. While tuning of expression levels by miRNAs is indeed occasionally referred to as an “epigenetic mechanism”, the reviewer strongly suggest to use a different more accurate term, such as “transcriptional regulation”. See e.g. PMID 29339796.

<https://www.nature.com/articles/nrm.2017.135.pdf>

We agree and changes “epigenetic mechanism” to “transcriptional regulation” throughout the revised text.

9. In Figure 1B it is suggested to also show effects of the other DHHCs beyond 5, 9 and 21 (i.e. Figure S2B). Why is #9 considered positive but #7 is not?

As correctly mentioned by the Reviewer, results of labelling experiments with radioactive 3H -palmitate shown in Supplementary Figure 2B demonstrate that overexpression of ZDHHC7 results in increase of 5-HT1AR palmitoylation similar to that obtained for the ZDHHC9. However, in case of ZDHHC7, this effect was mainly mediated by the increased palmitoylation of the lower 5-HT1AR protein band. In our previous study, we have shown that the double bands obtained for 5-HT1AR represent differently glycosylated receptor species (Gorinski et al. (2012) Mol Pharmacol.82:448-63). While the upper band contains processed carbohydrates and thus corresponds to receptors localized at the plasma membrane, the lower one represents receptors containing unprocessed carbohydrates of the high mannose type. This suggests that overexpression of the ZDHHC7 leads to the 5-HT1AR accumulation in Golgi and thus prevents its proper transport to the plasma membrane. In line with that, live-cell imaging analysis of the N1E cells overexpressing ZDHHC7-GFP and 5-HT1AR-mCherry demonstrated that most of the receptors were present in the intracellular compartments with only a minor fraction resided at the plasma membranes (Fig. R4). Based on these results, we decided not to focus on the role of ZDHHC7 in the 5-HT1AR palmitoylation.

Fig. R4. Subcellular distribution of GFP-tagged DHHC7 and mCherry-tagged 5-HT1AR in living N1E cells. Scale bar: 10 μ m. Line scans on the right show the intensity profiles for 5-HT1AR (red) and ZDHHCs (green). Grey bar shows the Golgi compartment.

10. Is the gel image in Figure 1E from a single blot or is it stitched together? There appear to be breaks e.g. between lanes 2 and 3 and between lanes 4 and 5 as well as 5 and 6 in the upper Palm 5-HT1AR row. Indeed one gets the impression that the –lanes have been obtained by covering the corresponding band or stitching different pieces together. This has indeed to be clarified by the authors showing the entire gel in e.g. supplementary documentation. After the careful inspection of the original images shown in Fig. 1E, we found no covering or stitching of protein bands. The small vertical lines visible on the right to Scr and DHHC5 lines as well as on the left to the DHHC21 line originates from the spill-over of samples during the gel loading. To further clarify this issue, we included in Fig. R5 below an image of the original gel used for the preparation of Fig. 1E as well as the same gel after oversaturation. The latter one clearly demonstrates that image shown in Fig. 1E originates from the single uncut gel.

We also would like to stress that all the images included in the manuscript were prepared strictly in accord with the Illustrations Processing Guide to Authors, and we never performed any selective covering, cropping or stitching of images. Needless to mention, we will be happy to run through all our illustrations together with the journal Editors to eliminate any potential doubts.

Fig. R5. Original uncut gel (on the left) and the same gel after artificial oversaturation (on the right) used for the creation of image shown in Fig. 1E. Gel image was assessed by Fusion SL bioluminescence image reader.

11. On page 5 it is stated “Noteworthy, knock-down of ZDHHC21 gave the more prominent reduction of 5-HT1AR palmitoylation, confirming this ZDHHC as a major palmitoyl-transferase for 5-HT1AR.” From Figure 1E it appears that DHH9 knock-down seems to result in the strongest effect.

We apologize for not being sufficiently clear here. As mentioned in our response to point 5, we introduced the following statement to the corresponding results section:

“Noteworthy, knock-down of ZDHHC9 and 21 gave the more prominent reduction of 5-HT1AR palmitoylation in comparison to the effect mediated by shRNA against ZDHHC5 (Fig. 1F), suggesting that both ZDHHC9 and ZDHHC21 represent relevant palmitoyl-acyltransferases (PATs) for 5-HT1AR. Based on the results obtained after ZDHHC overexpression (Fig. 1B and C), we decided to focus on ZDHHC21 as a more potent PAT for 5-HT1AR”.

Reviewer #2 (Remarks to the Author):

The manuscript by Gorinski et al showed that serotonin 1A receptor (5-HT1AR) is palmitoylated in the brain and its palmitoylation level is specifically reduced in the prefrontal cortex (PFC) in rodent models of depression and in the PFC of depressed suicide subjects. The authors identified ZDHHC21 as a major palmitoyl-transferase for 5-HT1AR and found the reduced expression of ZDHHC21 in the above rodent models and suicide subjects. Then, the authors demonstrated the causative relationship among reduced ZDHHC21 expression in the PFC, 5-HT1AR palmitoylation level, and depressive symptoms. This work proposes an interesting, new molecular mechanism for the pathogenesis of depressive symptoms. Addressing the following points would strengthen this paper.

Major comments

(1) The authors showed that knockdown of ZDHHC21 in PFC reduced the 5-HT1AR palmitoylation (Fig. 5C) and that a global protein palmitoylation was not affected by ZDHHC21 knockdown (Supplementary Fig. 7J). However, the silver staining presented in Supplementary Fig. 7J is not sufficient to neglect the involvement of palmitoylated proteins other than 5-HT1AR. Quantitative palmitome analysis combined with mass spectrometry is useful to confirm the specificity and their claim.

Based on the Reviewer's suggestion, we have verified substrate specificity of ZDHHC21 in the PFC using the quantitative palmitoylomics approach, which we have recently established in our lab (Sobocinska et al., 2018, Mol. Cell. Proteomics. 17:233-254). In particular, we used a high-throughput ABE proteomics approach that enables identification of S-Palmitoylated-Cys sites in complex biological mixtures. Using this unbiased approach in combination with the mass spectrometry-based protein identification to precisely ascertain the targets of S-palmitoylation, we compared palmitoylation profile in the PFC of control animals and those injected with scramble or shRNA against ZDHHC21. We specified inclusion criteria used for the selection of differential protein sets in detail in "Experimental procedures" section on pages 24-25. Overall, we observed more than 1.700 palmitoylated proteins. For data evaluation we applied an analytic approach developed for the global analysis of proteomics data (Fronslow et al., 2013, Nat Methods. 10:54-6).

Detailed analysis of our palmitoylomics results revealed that knock-down of ZDHHC21 indeed provoked a strong (app. 4-fold change plotted in log₂ scale, which correspond to app. 15 times decrease in palmitoylation level) and highly significant (p= 0.0028) decrease of 5-HT1AR palmitoylation in comparison to the scr samples (Fig. R6 below). Noteworthy that among three additional proteins, which palmitoylation was reduced to the similar extent, reduction of 5-HT1AR palmitoylation was at the highest level of confidence (p=0.0028), followed by proteosomal ubiquitin receptor ADRM1 (p=0.012), exopolyphosphatase PRUNE1 (p=0.017), STAT6 (p=0.017), and Hsc70-interacting protein (p=0.028). For two other proteins, including TRPC channel subfamily V (TRPV2) and phosphoglycerate mutase 1 (PGAM1), the level of confidence was comparable with that of the 5-HT1AR. However, for these proteins we obtained substantially lower decrease in palmitoylation (1.97-fold change for TRPV2 and 1.16-fold change for PGAM1). Interestingly, with the exception of 5-HT1AR, palmitoylation of all above mentioned proteins has not been reported before. More importantly, these proteins have not been reliably associated with major depressive disorder, suggesting that depression-like phenotype obtained after ZDHHC21 knock-down is likely to be mediated by the decreased 5-HT1AR palmitoylation.

Finally, we would like to thank the Reviewer for the motivating comment, which resulted in very interesting and highly related set of additional data. In the revised manuscript, we included the palmitoylomics results as an additional chapter into the "Results" section and presented these in Figure 5H and supplementary Tables 1A and 1B. In addition, palmitoylomics results are discussed on pages 15-16.

According to the Nature communications guidelines, we submitted our palmitoylomics raw data to the PRIDE repository (PXD012736).

At the current stage, raw data are only visible for Reviewers with the following account details:

USERNAME: reviewer55596@ebi.ac.uk

PASSWORD: ZhXXV6IV

In case of acceptance, these data will be available for all readers.

No.	UNIPROT ID	UniProt protein name	FOLD CHANGE shDHHC21/SCRAMBLE (LOG2(RATIO))	SIGNIFICANCE P-value t- test (ratio (shDHHC21/scramble)
	Q64264	5-hydroxytryptamine receptor 1A	-3.91	0,0028
1	Q9WTR1	Transient receptor potential cation channel subfamily V	-1.97	0,0022
2	Q9DBJ1	Phosphoglycerate mutase 1	-1,16	0,0028
3	Q9JKV1	Proteasomal ubiquitin receptor ADRM1	-2.11	0.0122
4	Q8BW1	Exopolyphosphatase PRUNE1	-6.64	0.0170
5	P52633	Signal transducer and transcription activator 6	-5.74	0.0172
6	Q99L47	Hsc70-interacting protein	-8.08	0,0282

Fig R6. Results of the comparative palmitoylomics analysis performed in the PFC of mice injected with scramble shRNA or shRNA against ZDHHC21. Volcano plot shows significance versus fold-change on the y and x axes, respectively. Vertical dotted grey lines mark 1-fold change plotted in log₂ scale which corresponds to 2 times decrease/increase in palmitoylation ratio. Less than 10% of all identified proteins were found to undergo such changes. Black dotted line indicate 2- fold change and less than 5% of all identified proteins were found to undergo such changes. Horizontal grey dotted lines indicate p values 0.05, and black dotted line – p=0.01.

(2) Knockdown of ZDHHC21 may have target off effects. Does loss of function mutant mouse of ZDHHC21 (used in Fig. 1F) display the similar depression-like behavior? As mentioned in our response to Reviewer#1 (point 7), non-conditional transgenic models often show strong compensatory effect during development. Therefore, we first analyzed whether the reduced 5-HT1AR palmitoylation obtained in the brains of newborn (P0) *Zdhhc21^{dep/dep}* animals would persist till adulthood. However, different from the *P0 Zdhhc21^{dep/dep}* animals, in the brains of adult (P30) mice no decrease in palmitoylation of 5-HT1AR could be observed (Fig. R3).

Thus, our results demonstrate that *Dep* mice can hardly be used as an appropriated model to verify our hypothesis about the role of 5-HT1AR palmitoylation in depression. I Also to note that no single publication of the five dealing with the *Dep* mouse line mentioned any depression-like behavioral abnormalities. This was further confirmed during our personal communication with Dr. Ian Smyth and Dr. Ian Jackson, who demonstrated that the *Dep* mouse contains a mutation in the *Zdhhc21* gen (PLoS Genet. 2009, 5:e1000748). Importantly, these mice display a spectrum of pathological symptoms not related to brain functions, including disrupted epidermal homeostasis (PLoS Genet. 2009, 5:e1000748), reduced vascular tone, which manifests in vivo as hypotension and tachycardia (Arterioscler Thromb Vasc Biol. 2016, 36:370-9) and endothelial dysfunction (Nat Commun. 2016, 7:12823). These deficits would further complicate usage of *Zdhhc21^{dep/dep}* mice as an eligible depression model.

In the revised manuscript, we added our results concerning 5-HT1AR palmitoylation in the brain of *P30 Dep* mouse to the Supplementary Figure 3E and discussed these data on page 6.

(3) The authors proposed that microRNA (miR-30e) might be involved in the regulation of ZDHHC21 expression as a negative regulator. To prove this hypothesis, the authors should examine whether miR-30e is increased in the rodent models of depression.

We would like to thank the Reviewer for this valuable suggestion. We decided to analyze the expression profile of miR-30a, miR-30e and miR-200a in human post-mortem mPFC samples, in order to directly evaluate the role of these micro-RNAs for depression.

These experiments revealed that the expression of miR30a and miR30e was significantly increased, while the expression of miR-200a was drastically reduced in samples from depressive suicide victim (Fig. R1). These new results provide a strong experimental support for our hypothesis on the role of defined micro-RNAs in the development of depression, and we thank the reviewer for this important comment. The micro-RNA expression data now shown in the revised Figure 6F and mentioned on pages 13, 16-17, and 26. In the future studies, we are planning to evaluate miR-30a, -30e, and -200a as potential biomarkers to identify MDD patients with suicidal thoughts.

Minor comments

(1) In the abstract, the second sentence “Here we demonstrated that 5-HT1AR is palmitoylation in human and rodent brains” should be “Here we demonstrated that 5-HT1AR is palmitoylated in human and rodent brains”.

We have corrected this mistake in the revised manuscript

(2) On page 7, line 7 from the bottom, “Supplementary Fig. 6” should be “Supplementary Fig. 5E”. As one of general readers, I suggest that the description/citation of Supplementary Figure should be specifically indicated throughout the manuscript, such as “Supplementary Fig. 5E”, but not “Supplementary Fig. 5”.

We have corrected the description of Supplementary Fig. 5 in the revised text. We also specified description of all Supplementary figures according to Reviewer’s suggestion and thank Reviewer for this suggestion.

(3) In the supplementary figures, on page 9, line 3 from the bottom, (E) is (J).

This is corrected in the revised manuscript

Reviewer #3 (Remarks to the Author):

This study examines palmitoyl-transferase ZDHHC21 in 5-HT1A palmitoylation and depression phenotypes. Depletion of ZDHHC21 reduces 5-HT1A palmitoylation and signaling, and both are reduced in depressed PFC. Knock-down of ZDHHC21 in PFC results in depression-related behavior. miR30e is identified as a negative regulator of ZDHHC21. The authors present extensive data supporting the effects of ZDHHC9 and 21 in 5-HT1A palmitoylation, and a partial inhibition of 5-HT1A signaling in transfected cells, with a striking effect on 5-HT1A-GIRK coupling in hippocampal neurons. Overall their data support the argument that reduced ZDHHC21 in PFC could mediate a depression phenotype, but the exact role of 5-HT1A palmitoylation is unclear. Furthermore the role of miRNA in stress-induced alterations is unclear.

Since ZDHHC21 may have many palmitoylation targets (e.g., see Kang R et al., Nature 456: 904, 2008), to show that 5-HT1A palmitoylation is critical for the depression phenotype following its depletion, studies need to be done on a 5-HT1A $-/-$ background to block the effect of ZDHHC21 depletion. Alternately, the effect of knock-in or rescue of a palmitoylation-defective 5-HT1A mutant could be tested.

We agree that defined ZDHHC can have multiple substrates, depending on site and time of expression. To verify the substrate specificity of ZDHHC21 in the PFC, we first applied the quantitative palmitoylomics approach, which we have recently established in our lab (Sobocinska

et al., 2018, *Mol. Cell. Proteomics*. 17:233-254). Using this technique, we compared palmitoylation profile in the PFC of control animals and those injected with scramble or shRNA against ZDHHC21.

Detailed analysis of our palmitoylomics results revealed that knock-down of ZDHHC21 indeed provoked a strong (app. 4-fold change plotted in \log_2 scale, which correspond to app. 15 times decrease in palmitoylation level) and highly significant ($p=0.0028$) decrease of 5-HT1AR palmitoylation in comparison to the scr samples (Fig. R6). Noteworthy that among three additional proteins, which palmitoylation was reduced to the similar extent, reduction of 5-HT1AR palmitoylation was at the highest level of confidence ($p=0.0028$), followed by proteosomal ubiquitin receptor ADRM1 ($p=0.012$), exopolyphosphatase PRUNE1 ($p=0.017$), STAT6 ($p=0.017$), and Hsc70-interacting protein ($p=0.028$). For two other proteins, including TRPC channel subfamily V (TRPV2) and phosphoglycerate mutase 1 (PGAM1), the level of confidence was comparable with that of the 5-HT1AR. However, for these proteins we obtained substantially lower decrease in palmitoylation (1.97-fold change for TRPV2 and 1.16-fold change for PGAM1). Interestingly, with the exception of 5-HT1AR, palmitoylation of all above mentioned proteins has not been reported before. More importantly, these proteins have not been reliably associated with major depressive disorder, suggesting that depression-like phenotype obtained after ZDHHC21 knock-down is likely to be mediated by the decreased 5-HT1AR palmitoylation.

Though the above-mentioned proteins have not been associated with depression, we agree that their palmitoylation might theoretically have an impact on the depressive phenotype. Thus, we followed Reviewer's suggestion and developed experimental strategy to directly verify the role of ZDHHC21-mediated 5-HT1AR palmitoylation for the depression phenotype. To this end, we have knocked-down ZDHHC21 in the PFC using the AAV construct encoding the shRNAs. After 4 weeks, when animals started to develop depression-like phenotype, we have done a single injection of the 5-HT1AR selective agonist 8-OH-DPAT. We chose this strategy, as multiple prior studies evidently show that acute 8-OH-DPAT injection results in a strong 5-HT1AR-mediated anti-depressive effects in WT animals as assessed by decreased immobility time in the forced swimming test (e.g. Borsini (1995) *Neurosci Biobehav Rev*; O'Neill and Conway (2001) *Neuropsychopharmacology*; Sugimoto et al (2010) *Eur J Pharmacol*; Miyake et al (2014) *Pharmacol Biochem Behav*). We were able to reproduce this 5-HT1AR-mediated effect of 8-OH-DPAT in control conditions, when DHHC21 was normally expressed (Fig. R7 below). However, acute injection of 8-OH-DPAT failed to induce any anti-depressive effect in animals after knocked-down of DHHC21, which results in significant reduction of 5-HT1AR palmitoylation in the PFC and, consequently, in impaired receptor functions (Fig. R7). In combination with results of our palmitoylomics experiments, this data provides a direct evidence for ZDHHC21-mediated 5-HT1AR palmitoylation playing a central role in the etiology of depressive phenotype.

Fig. R7. Relative changes of immobility time in the forced swim test (FST) measured in 3-month-old C57BL6/J male control mice or in animals 30 days after administration of AAV encoding for shRNA against ZDHHC21. FST was carried out 20 min after the single injection of the selective 5-HT1AR agonist 8-OH-DPAT (8-OH, 1mg/kg, i.p.). Statistical significance between values is noted (* $P < 0.05$, one-way ANOVA), $n > 7$ mice per condition.

We also believe that this approach is preferable over using full 5-HT1AR KO mice, because (i) KO animals seem to develop compensatory effects leading to the only mild increase of anxiety behavior without any depression-like phenotype (Parks et al. (1998) PNAS, 95:10734-9, Heisler et al (19989 PNAS, 95:15049-54), (ii) in KO mice, the effects of the 5-HT1AR palmitoylation in the PFC (which is the main focus of our study) could be “contaminated” by the absence of the pre-synaptic receptors.

Moreover, we have made the necessary efforts to follow the other Reviewer’s suggestion to perform rescue experiments with non-palmitoylated 5-HT1AR mutant.

To this end, we generated bi-cistronic adenovirus associated viral (AAV) constructs encoding shRNAs to silence the endogenously expressed 5-HT1AR and to simultaneously express the shRNA-resistant, eGFP-tagged WT or palmitoylation-deficient 5-HT1AR under control of the synapsin promoter. These AAV constructs were successfully evaluated in cultured neurons (Fig. R8A below). However, stereotactic AAV injections into the mouse PFC revealed that even after a long post-infection time (4 to 6 weeks), amounts of the endogenous receptors were only moderately affected (Fig. R8B), though recombinant protein was still expressed (Fig. R8B and C). This demonstrates that the endogenous post-synaptic 5-HT1AR protein is very stable in vivo. This is in line with previous observations, showing that the post-synaptic 5-HT1AR is highly resistant against desensitization and/or degradation under basal conditions (Blier and de Montigny (1994) Trends Pharmacol Sci.). In addition, our own results demonstrated that, in contrast to the 5-HT7R, prolonged stimulation of cells expressing 5-HT1AR with serotonin showed no significant receptor internalization (Renner et al (2012) J Cell Sci). Thus, due to intrinsic properties of the post-synaptic 5-HT1AR proteins, it seems to be technically impossible to successfully perform knock-down/rescue experiments in the PFC in the way suggested by Reviewer#3.

Fig. R8. Replacement of the endogenous (end.) 5-HT1AR through the shRNA-resistant 5-HT1AR-GFP WT or palmitoylation deficient mutant (A) in primary cultures of cortical neurons and (B) in the mouse prefrontal cortex (PFC) 30 days after bilateral stereotactic injections of the corresponding AAVs. Representative Western blots are shown. (C) Microscopic images of the mouse brain area injected with AAV construct encoding for shRNA against 5-HT1AR and shRNA-resistant, palmitoylation-deficient 5-HT1AR-eGFP mutant. DAPI and GFP channels are shown.

Finally, we would like to thank the Reviewer for the motivating comment, which resulted in very interesting and highly related set of additional data. In the revised manuscript, we included the palmitoylomics and 8-OH-DPAT data as an additional chapter into the “Results” section on pages 11-12 and presented these in Figure 5H, supplementary Tables 1A and 1B, and supplementary figure 7N. In addition, these results are discussed on pages 15-16.

According to the Nature communications guidelines, we submitted our palmitoylomics raw data to the PRIDE repository (PXD012736).

At the current stage, raw data are only visible for Reviewers with the following account details:

USERNAME: reviewer55596@ebi.ac.uk

PASSWORD: ZhXXV6IV

In case of acceptance, these data will be available for all readers.

Several minor grammatical errors need correction: e.g., Abstract, use present tense: Here we demonstrate that 5-HT1AR is palmitoylated... and identify ZDHHC21...; avoid run-on sentences (e.g. the above sentence could be split).

We agree with the Reviewer and therefore made efforts to ensure that proof reading of the revised manuscript is done carefully.

Specific comments:

1. Introduction: it is important to note that over-expression of 5-HT1A in Gross et al. was done in pyramidal cells and in early post-natal mice.

We corrected the statement relevant to the Gross et al. citation on page 3 as follows:

“The over-expression of the 5-HT1AR induced during the early postnatal period in the forebrain, but not in the raphe nuclei, has been found to be sufficient to rescue the behavioral phenotype of the knockout mice. This data suggests an important role of postsynaptic 5-HT1AR in depressive disorders.”

2. Results, p. 4: in the screen of PTs, ZDHHC5, 9, and 21 enhanced 5-HT1A palmitoylation: are these PTs conserved, do they form a sub-family?

ZDHHC5, 9 and 21 belong to the three different subfamilies of ZDHHC proteins with a relative low sequence homology. Sequence homology between human ZDHHC5 and 9 is 43%, between ZDHHC21 and 9 - 29%, and between ZDHHC21 and 5 - 26%. Percentages of homology in mouse and rat are quite similar. On the other hand, each of these ZDHHCs is highly conserved between different mammalian species (with app. 98% homology between mouse, rat and human isoforms). We added this information to the “Results” section on pages 4-5.

3. Supp. Fig. 2B: What are the #; show error bars; what was the N-value?

“#” depicts ZDHHCs with a more prominent effect on 5-HTAR palmitoylation. To avoid misunderstanding, we have replaced it by arrows in the revised version. Figure legends was corrected accordingly.

Experiments with the radioactive ³H-palmitate labelling were repeated two times. Labelling with a radioactive palmitate represents a simple and straightforward method to identify candidate PAT:substrate pairs (Fukata et al., Methods, 2006, 40: 177–182; Tsutsumi et al., 2009, Mol Cell Biol. 29:435-47). ZDHHCs, whose overexpression resulted in a largest increase of ³H-palmitate incorporation, were then selected for the systematic quantitative analysis shown in Fig. 1.

ZDHHC5 seems to have the greatest effect on 5-HT1A palmitoylation, was this consistent or significant? Some of the other ZDHHCs seem to have activity, why were they excluded?

What was the rationale for focusing on ZDHHC21?

We apologize for not being sufficiently clear here. The rationality for our focus on ZDHHC21 was based on following observations:

(i) Our quantitative analysis shown in Figure 1C demonstrates that only over-expression of ZDHHC21 results in significantly increased 5-HT1AR palmitoylation.

(ii) When expression of ZDHHC5, 9 and 21 was evaluated in human samples obtained from depressive suicide victims as well as in both rodent depression models, we found that only expression of ZDHHC21 was significantly reduced throughout all the depression models.

Indeed, in the human MDD samples, ZDHHC9 and 21 were affected, in the mouse model – only 21, and in the rat model ZDHHC5, 9 and 21.

We also quantified the effects of shRNAs against ZDHHC5, 9 and 21 on 5-HT1AR palmitoylation shown in Fig. 1E and F. In case of ZDHHC5 knock-down, we obtained a slight decrease in 5-HT1AR palmitoylation ($p = 0.049$). Knock-down of either ZDHHC9 or ZDHHC21 resulted in a more significant decreased receptor palmitoylation ($p=0.006$ for ZDHHC9 and $p=0.008$ for ZDHHC21 knock-down), suggesting that ZDHHC9 might also be involved in 5-HT1AR palmitoylation.

We hope that these new results in combination with above mentioned observations clarify our decision to focus on ZDHHC21 as an obviously more potent 5-HT1AR acyl-transferase. To clarify this point, we have now included the following statement to the results section:

“Noteworthy, knock-down of ZDHHC9 and 21 gave the more prominent reduction of 5-HT1AR palmitoylation in comparison to the effect mediated by shRNA against ZDHHC5 (Fig. 1F), suggesting that both ZDHHC9 and ZDHHC21 represent relevant palmitoyl-acyltransferases (PATs) for 5-HT1AR. Based on the results obtained after ZDHHC overexpression (Fig. 1B and C), we decided to focus on ZDHHC21 as a more potent PAT for the 5-HT1AR”.

4. Supp. Fig. 2D: ZDHHC9/21 appear to relocate 5-HT1A to Golgi, while ZDHHC5 appears to maintain its localization to PM. Quantify.

In line with the Reviewer’s suggestion we have quantified the distribution of 5-HT1AR after co-expression with ZDHHC5, -9, or -21 (Fig. R9 below). These experiments revealed a preferential plasma membrane localization of the 5-HT1AR after co-expression with all the relevant ZDHHCs. We added this data to the revised manuscript as Supplementary Fig. 2D and E and briefly mentioned on page 5.

Fig. R9. Upper panel: subcellular distribution of GFP-tagged DHHC5, -9 and -21 and mCherry-tagged 5-HT1AR in living N1E cells. Scale bars: 20 μ m. Line scans on the right show

intensity profiles for 5-HT1AR (red) and ZDHHCs (green). Grey bar shows Golgi area. Lower panel: quantification of the intracellular distribution of 5-HT1AR and indicated ZDHHCs. N=3 biological replicates. In each experiment, at least 5 cells were analyzed.

5. Results, p. 5: In terms of signaling, G α -i subunits are palmitoylated, which is crucial for receptor-G protein signaling (e.g., see Schattauer SS, Nature Comm 2017); did depletion of ZDHHCs affect G α palmitoylation?

We have performed these experiments, but did not detect any differences in G α /o palmitoylation in the PFC of mice after injection of shRNA against ZDHHC21 (Fig. R10 below). Moreover, results of our palmitoylomics analysis in the mouse brain did not show any changes in G α /o palmitoylation in the PFC after ZDHHC21 knock-down (change in the palmitoylation ratio between shDHHC21 and scr samples is 0.92, p=0.8). Taken together, these results suggest that ZDHHC21 is not involved in G α /o palmitoylation. This is also in line with the reported findings by Tsutsumi and colleagues, who identified ZDHHC3 and ZDHHC7 as main PATs for G α (Tsutsumi et al., 2009, Mol Cell Biol. 29:435-47)

Fig. R10. PFCs of mice were isolated 30 days after injection either with vehicle or scrambled shRNA, or shRNA against ZDHHC21 and subjected to ABE assay to define palmitoylation of the G α (i) subunit.

6. Fig. 1: in fig. 1b there seems to be endogenous palmitoylation of 5-HT1A, yet in 1d no palmitoylation was detected in the “mix” samples: explain. What post-test was used?

We appreciate the opportunity to clarify this point, as here is a straightforward explanation. In Fig. 1D we analyzed interaction between receptor and relevant ZDHHCs rather than 5-HT1AR palmitoylation. In the revised manuscript, we now explain this issue in detail on page 5. It now reads:

“Direct evidence for interaction between 5-HT1AR and ZDHHC5, -9, and -21 was provided in co-immunoprecipitation experiments (Figure 1D). In these experiments specific interaction between 5-HT1AR and relevant ZDHHCs was analyzed by co-immunoprecipitation experiments in N1E-115 cells co-expressing haemagglutinin (HA)-tagged 5-HT1AR and GFP-tagged ZDHHCs. Figure 1D shows that after immunoprecipitation with an antibody against the GFP-tag, the HA-tagged receptor could be identified only in samples derived from cells co-expressing both HA- and GFP-tagged proteins. To assay the extent of artificial protein aggregation, cells expressing only one type of protein (i.e. either HA-5-HT1AR or GFP-tagged ZDHHC) were mixed prior to lysis and analyzed in parallel (“mix” samples). As shown in Figure 1D, both 5-HT1AR and ZDHHC can be detected by the corresponding antibody (visible in “input” fraction), but no co-immunoprecipitation was observed. This further verifies the specificity of 5-HT1AR-ZDHHC interaction.”

7. Fig. 2C: define relative cAMP amplitude: relative to pretreatment?

We appreciate this comment and corrected the legend to figure 2C as follows:

“(B) Graphs show activation time constant and (C) changes of cAMP response amplitude relative to pretreatment.”

8. Fig. 3, 4: show timeline for stressors and provide more details for the sequence and frequency of stressors, especially in mice.

We added a more detailed description for both mouse and rat depression models to the “Experimental Procedures” section on pages 19 and 18, respectively.

Quantify levels of Total 5-HT1A receptors: was the reduction in palmitoylated 5-HT1A simply reflecting a reduction in total 5-HT1A?

We quantified the expression levels of 5-HT1AR in the PFC of mouse and rat depression models (Supplementary Fig. 5A and Supplementary Fig. 6A) as well as in the PFC of mice after knock-down of ZDHHC21 by shRNA (Fig. 5E). In all these experiments we obtained no significant differences of the 5-HT1AR expression. In addition, we compared the expression of 5-HT1AR in post-mortem human samples from control and suicide groups but found no significant differences between the groups. These data are included in the revised Supplementary Fig. 8A.

What is the evidence of the specificity of the 5-HT1A antibody for 5-HT1A (e.g., is there any 5-HT1A staining in 5-HT1A knockout mice)?

To test for the antibody specificity, we performed Western blot analysis as well as immunofluorescent staining in cultured hippocampal neurons after the application of selective blocking peptide (#ASR-021, Alomone Labs) (Fig. R11 below). Results of these experiments demonstrate that the staining with anti-5-HT1AR antibody disappears after pre-treatment with the blocking peptide.

Fig. R11. Test for the specificity of the antibody against 5-HT1AR. (A) Brain lysates from mouse and rat were subjected to the Western blot analysis with anti-5-HT1AR antibody pre-treated with (left) or without (right) blocking peptide. (B) Primary hippocampal neurons were infected at DIV6 with vector encoding for RFP. Neurons were fixed on DIV15, incubated with anti-5-HT1AR receptor antibody (two upper panel) pre-treated with or without blocking

peptide. In lower panel neurons were incubated only with the secondary antibody. Images were analyzed for expression of endogenous 5-HT1A using confocal microscopy. Maximum intensity projection images are shown.

9. Fig. 5B: How large was the spread of infection: did it cover the entire PFC, were other cortical or hippocampal areas infected? The protocol needs to be described in detail.

We added an extended protocol concerning the stereotactic injection to “Experimental Procedures” section on page 20. We also included new images as Supplementary Fig.7E demonstrating specific AAV infection within the PFC.

10. Fig. 6: In the postmortem samples, we the expression of 5-HT1A differing in control vs. suicide PFC?

We have quantified the expression of 5-HT1A in PFC in post-mortem samples but found no significant differences between control and suicide groups. These data are included in the revised Supplementary Fig. 8A.

11. Figs. 3, 4, 6: Is miR30e or 30a altered in depression? What is the effect of blocking miR30e on stress induced depression?

In line with the Reviewer’s suggestion, we analyzed the expression profiles of miR-30a, miR-30e and miR-200a. In order to directly evaluate the role of these micro-RNAs for depression, we decided to use human post-mortem mPFC samples. These experiments revealed that the expression of miR-30a and miR-30e was significantly increased, while the expression of miR-200a was severely attenuated in samples from depressive suicide victim (Fig. R1). These new results thus provide a strong experimental support for our hypothesis concerning the role of defined micro-RNAs in development of depression, and we thank Reviewer for such important comment. The data on the micro-RNA expression are summarized in revised Figure 6F and mentioned on pages 13, 16-17, and 26. In the future studies, we are planning to evaluate miR-30a, -30e, and -200a as potential biomarkers to identify MDD patients with suicidal thoughts.

12. Discussion: It is important to emphasize that studies showing depression associated with increased 5-HT1A were in the raphe (autoreceptors).

We thank the Reviewer for this comment. In the revised manuscript we have extended the corresponding part of the “Discussion” section on page 14 by the statement:

“In another study, however, analysis of the post-mortem brains of depressed subjects in comparison with control samples revealed a specific upregulation of 5-HT1A autoreceptors in the raphe area, with no changes in postsynaptic 5-HT1A receptor sites (Stockmeier CA, Shapiro LA, Dilley GE, Kolli TN, Friedman L, Rajkowska G. Increase in serotonin-1A autoreceptors in the midbrain of suicide victims with major depression-postmortem evidence for decreased serotonin activity. J Neurosci 1998; 18: 7394-401)”.

Note one study that did show impaired 5-HT1A signaling in depressed suicides should be mentioned (Hsiung SC et al., J. Neurochem. 2003).

We carefully inspected our reference list and found that the study by Hsiung and colleagues (Hsiung, S. et al. Attenuated 5-HT1A receptor signaling in brains of suicide victims: involvement of adenylyl cyclase, phosphatidylinositol 3-kinase, Akt and mitogen-activated protein kinase. J. Neurochem. 87, 182–194 (2003)) was already cited in our manuscript under the reference number 41.

13. Discussion: The data in Fig. 3 indicates that a 5-HT1A receptor lacking palmitoylation sites can still couple to cAMP or ERK1/2, hence palmitoylation is not essential for coupling but enhances coupling, and does not down-regulate coupling, but partially attenuates it.

In line with the Reviewer comment we corrected the corresponding statement on pages 3-4 as follows:

“Previously, we showed that 5-HT1AR is palmitoylated at its C-terminal cysteine residues Cys417 and Cys420 and that the mutation of palmitoylated cysteines declines G_i-protein receptor-mediated signaling via mislocalization of receptor outside of the lipid rafts^{13,14}. We also found that 5-HT1AR palmitoylation efficiency was not modulated by the receptor stimulation with agonists¹³”.

14. Discussion: The data do not resolve whether the effect of reduced ZDHHC21 on depression is mediated solely by reduced 5-HT1A palmitoylation.

In the revised manuscript, we discussed this issue in detail on page 15, as explained below.

“Results of our palmitoylomics experiment revealed 5-HT1AR as a main substrate for ZDHHC21 in the PFC. Although we cannot completely exclude an impact of other proteins, whose palmitoylation was affected under ZDHHC21 knock-down (e.g. ADRM1, PRUNE1, STAT6, Hsc70-interacting protein, TRPV2, PGAM1), these proteins have not been reliably associated with major depressive disorder. This suggests that depression-like phenotype obtained after ZDHHC21 knock-down is likely to be mediated by the decreased 5-HT1AR palmitoylation. In combination with results obtained after acute injection of 5-HT1AR agonist 8-OH-DPAT, which failed to evoke any anti-depressive effects in animals after knock down of DHHC21, this data provides a direct evidence for ZDHHC21-mediated 5-HT1AR palmitoylation playing a central role in the etiology of depressive phenotype.

Noteworthy, even the substrate specificity represents a critical component determining functional consequences of DHHC activity, this issue was not yet systematically investigated for the other ZDHHC members. In majority of ZDHHC-related studies, including identification of DHHCs responsible for palmitoylation of phospholemman (Howie et al., 2014, PNAS), melanocortin receptor MC1R (Chen et al., 2016, Science) and CD36 (Wang et al., 2019, Cell Report), the question about the substrate specificity has not been addressed in details”.

REVIEWERS' COMMENTS:

Reviewer #1 (Remarks to the Author):

The interesting new findings in this paper are:

1. ZHHC21 PFC Knockdown abolishes the antidepressant-like effect of 8OH-DPAT (5HT1A agonist)
2. miR30e is overexpressed in the PFC of suicide victims
3. ZDHHC21 overexpression in the 5-HT1A palmitoylation in the PFC and reduces depressive like behaviour in mice

The results of the study indicate that MiR30e -> ZHHC21 -> 5HT1A palmitoylation is a valid drug target and that dysregulation correlates

In relation to previous criticism raised by this reviewer the authors have done extensive additional experiments in response to the previous criticism:

- They extensively characterized the ZDHHC21 KD animals performing a series of novel behavioural tests. Using the FST test they found that the only got augmentation of depression in the KD ZDHHC21 animals:
- They quantified the palmitoylation in Fig 5C.
- They did additional experiments to show that the 5-HT1AR expression is unchanged.
- They analysed the acute effect of 8-OH-DPAT on the KD animals
- They found that miR30a and miR30e were induced and miR-200a reduced in post mortem mPFC samples from depressive suicide victims.
- They extended the demographic data.. The reviewer think these data should be shown in the supplementary data.
- They experimentally motivated their focus on ZDHHC21
- They overexpressed ZDHHC21 in the PFC of mice subjected to the short day conditions and found increase of 5-HT1A palmitoylation in the PFC which is an important finding.
- In addition, they corrected a number of additional errors/ambiguities.

In conclusion, the manuscript has now been extensively improved.

Reviewer #2 (Remarks to the Author):

The authors sincerely addressed all my comments. I am satisfied with their response.

Reviewer #3 (Remarks to the Author):

This study examines palmitoyl-transferase ZDHHC21 in 5-HT1A palmitoylation and depression phenotypes. Depletion of ZDHHC21 reduces 5-HT1A palmitoylation and signaling, and both are reduced in depressed PFC. Knock-down of ZDHHC21 in PFC results in depression-related behavior. miR30e is identified as a negative regulator of ZDHHC21.

The authors present extensive data supporting the effects of ZDHHC9 and 21 in 5-HT1A palmitoylation, and a partial inhibition of 5-HT1A signaling in transfected cells, with a striking effect on 5-HT1A-GIRK coupling in hippocampal neurons. Overall their data support the argument that reduced ZDHHC21 in PFC is implicated in a depression phenotype. The findings are novel and for the first time implicate receptor palmitoylation and altered regulation of palmitoylases in depression.

The authors have addressed my previous concerns with new data that greatly strengthen the conclusions of the study.

Some issues remain:

1. Abstract: miRNA, which reduces translation and/or mRNA stability, is generally thought to act at the post-transcriptional (rather than transcriptional) level; however its up-regulation in depressed subjects may be a transcriptional regulation, although this was not tested. Hence the authors need to be more careful in referring to transcriptional vs. post-transcriptional events.
2. Introduction: Some typos include data suggest (data is plural); reduces rather than declines;
3. Results: p. 5; sequence homology should be specified as amino acid sequence homology.
4. Results, p. 10: Discuss why the TST did not show differences upon knockdown of ZDHHC21 in PFC, yet the FST, which is another behavioral despair assay, did.
5. Results, p. 11: 1,737 palmitoylated proteins
6. Results, p. 12: In the DPAT experiment, specify where ZDHHC21 was knocked-down (PFC?).
7. Results, p. 12: I think it would be worth including the R1 Table of subject demographics, since it provides post-mortem interval. In addition, the RIN values should be included to assure that the RNA quality is similar between groups. If anonymity is a concern, the cause of death could be omitted.
8. Table R1: Several of the MDD were on medication; can the authors rule out that the increase in miRNA or decrease in ZDHHCs is due to medication? Otherwise this needs to be stated as a caveat.
9. Supp. Fig. 2E: Indicate statistical significance (\pm DHHC).

REVIEWERS' COMMENTS:

Reviewer #1 (Remarks to the Author):

The interesting new findings in this paper are:

1. ZHHC21 PFC Knockdown abolishes the antidepressant-like effect of 8OH-DPAT (5HT1A agonist)
2. miR30e is overexpressed in the PFC of suicide victims
3. ZDHHC21 overexpression in the 5-HT1A palmitoylation in the PFC and reduces depressive like behaviour in mice

The results of the study indicate that MiR30e → ZHHC21 → 5HT1A palmitoylation is a valid drug target and that dysregulation correlates

In relation to previous criticism raised by this reviewer the authors have done extensive additional experiments in response to the previous criticism:

- They extensively characterized the ZDHHC21 KD animals performing a series of novel behavioural tests. Using the FST test they found that they only got augmentation of depression in the KD ZDHHC21 animals:
- They quantified the palmitoylation in Fig 5C.
- They did additional experiments to show that the 5-HT1AR expression is unchanged.
- They analysed the acute effect of 8-OH-DPAT on the KD animals
- They found that miR30a and miR30e were induced and miR-200a reduced in post mortem mPFC samples from depressive suicide victims.
- They extended the demographic data. The reviewer think these data should be shown in the supplementary data.
- They experimentally motivated their focus on ZDHHC21
- They overexpressed ZDHHC21 in the PFC of mice subjected to the short day conditions and found increase of 5-HT1A palmitoylation in the PFC which is an important finding.
- In addition, they corrected a number of additional errors/ambiguities.

In conclusion, the manuscript has now been extensively improved.

Reply:

We thank the Reviewer for such positive evaluation! On editorial advice, we excluded extended demographic data table from the final version of manuscript because of ethical considerations. This data is confidentiality issue because the ages and exact cause of death are given.

Reviewer #2 (Remarks to the Author):

The authors sincerely addressed all my comments. I am satisfied with their response.

Reply:

We thank the Reviewer for the positive evaluation!

Reviewer #3 (Remarks to the Author):

This study examines palmitoyl-transferase ZDHHC21 in 5-HT1A palmitoylation and depression phenotypes. Depletion of ZDHHC21 reduces 5-HT1A palmitoylation and signaling, and both are reduced in depressed PFC. Knock-down of ZDHHC21 in PFC results in depression-related behavior. miR30e is identified as a negative regulator of ZDHHC21. The authors present extensive data supporting the effects of ZDHHC9 and 21 in 5-HT1A palmitoylation, and a partial inhibition of 5-HT1A signaling in transfected cells, with a striking effect on 5-HT1A-GIRK coupling in hippocampal neurons. Overall their data support the argument that reduced ZDHHC21 in PFC is implicated in a depression phenotype. The findings are novel and for the first time implicate receptor palmitoylation and altered

regulation of palmitoylases in depression.

The authors have addressed my previous concerns with new data that greatly strengthen the conclusions of the study.

Some issues remain:

1. Abstract: miRNA, which reduces translation and/or mRNA stability, is generally thought to act at the post-transcriptional (rather than transcriptional) level; however its up-regulation in depressed subjects may be a transcriptional regulation, although this was not tested. Hence the authors need to be more careful in referring to transcriptional vs. post-transcriptional events.

Reply:

Based on Reviewer's comment and editorial suggestion, we have excluded term "transcriptional" from abstract. As it was correctly mentioned by the Reviewer, this issue needs more detailed investigation.

2. Introduction: Some typos include data suggest (data is plural); reduces rather than declines;

Reply:

We have corrected typos accordingly.

3. Results: p. 5; sequence homology should be specified as amino acid sequence homology.

Reply:

We thank the Reviewer for this suggestion. In the revised manuscript, we have replaced "sequence homology" by "amino acid sequence homology".

4. Results, p. 10: Discuss why the TST did not show differences upon knockdown of ZDHHC21 in PFC, yet the FST, which is another behavioral despair assay, did.

Reply:

We apologize for not being sufficiently clear here. In the revised manuscript, we introduced the following statement to the corresponding results section on p. 10:

"The lack of visible effect in the tail suspension test, which is often used for analysis of the depression-like behaviour in rodent, can be explained by the fact that the majority of the C57BL/6 mice tested in this paradigm climbed up their tails during the test session. Such behaviour is specific for the C57BL/6 mouse line, which is in accordance with previous observations (Mayorga and Lucki (2001) Limitations on the use of the C57BL/6 mouse in the tail suspension test. Psychopharmacology 155, 110–112)".

5. Results, p. 11: 1,737 palmitoylated proteins

Reply:

This typo was corrected.

6. Results, p. 12: In the DPAT experiment, specify where ZDHHC21 was knocked-down (PFC?).

Reply:

As it is correctly mentioned by the Reviewer, in 8-OH-DPAT experiments we knocked-down ZDHHC21 in the PFC. In the revised version we added this information to the corresponding results part.

7. Results, p. 12: I think it would be worth including the R1 Table of subject demographics, since it provides post-mortem interval. In addition, the RIN values should be included to assure that the RNA quality is similar between groups. If anonymity is a concern, the cause of death could be omitted.

Reply:

We fully agree with the Reviewer at this point. On editorial advice, we excluded extended demographic data table from the final version of manuscript because of ethical considerations. This data is confidentiality issue because the ages and exact cause of death are given.

8. Table R1: Several of the MDD were on medication; can the authors rule out that the increase in miRNA or decrease in ZDHHCs is due to medication? Otherwise this needs to be stated as a caveat.

Reply:

We thank the Reviewer for this important note. In order to test for potential role of medication in MDD individuals that died by suicide, we compare data points distribution obtained for miRNA and ZDHHC21 expression in these individuals and control subjects. Shapiro-Wilk test, which is known to provide the best power for analysis, revealed that the data point distribution in both groups (i.e. combined MDD individuals and control) undergoes normal distribution. Similar results were also obtained with Anderson-Darling test. These results suggest that previous medication seems to be not involved in regulation of miRNA and ZDHHC expression.

9. Supp. Fig. 2E: Indicate statistical significance (\pm DHHC).

Reply

In line with the Reviewer's suggestion, we have indicated statistical significance in Supplementary Figure 2E.